# DEEP LITERATURE SURVEY AUTOMATION WITH AN ITERATIVE WORKFLOW

## ABSTRACT

Automatic literature survey generation has attracted increasing attention, yet most existing systems follow a one-shot paradigm, where a large set of papers is retrieved at once and a static outline is generated before drafting. This design often leads to noisy retrieval, fragmented structures, and context overload, ultimately limiting survey quality. Inspired by the iterative reading process of human researchers, we propose IterSurvey, a framework based on recurrent outline generation, in which a planning agent incrementally retrieves, reads, and updates the outline to ensure both exploration and coherence. To provide faithful paper-level grounding, we design paper cards that distill each paper into its contributions, methods, and findings, and introduce a review-and-refine loop with visualization enhancement to improve textual flow and integrate multimodal elements such as figures and tables. Experiments on both established and emerging topics show that IterSurvey substantially outperforms state-of-the-art baselines in content coverage, structural coherence, and citation quality, while producing more accessible and better-organized surveys. To provide a more reliable assessment of such improvements, we further introduce Survey-Arena, a pairwise benchmark that complements absolute scoring and more clearly positions machine-generated surveys relative to human-written ones.

## 1 INTRODUCTION

Automatic literature survey generation has recently attracted growing attention due to its potential to help researchers quickly grasp new domains, identify key trends, and reduce the burden of manual reviews. Following Wang et al. (2024b), current systems generally adopt a multistage pipeline (Liang et al., 2025; Yan et al., 2025; Wang et al., 2025): The process begins with a topic description, usually consisting of a few tokens, which is directly used to retrieve a large collection of candidate papers. Due to the context window limitation of large language models (LLMs), the retrieved papers are divided into multiple groups, for each, an LLM agent generates a survey section outline based on the corresponding subset of papers. These group-level outlines are subsequently merged into a global draft outline. Once the draft outline is obtained, the system performs section-wise retrieval to collect references for section writing and then generates the corresponding text passages. Finally, a global review and integration process is applied, in which the drafted survey is iteratively polished to improve readability and overall consistency.

The above approach takes a "one-shot" planning paradigm, retrieves a comprehensive set of papers and construct a global outline from a single, static starting point. However, this approach lacks a structured understanding of individual papers, relying only on high-level signals to construct the outline. As a result, it becomes brittle, especially when applied to complex or emerging domains where nuanced and evolving information is crucial. This limitation leads to several challenges: **First, retrieval can be imprecise and static** due to reliance on a short topic description (often just a few tokens) as the retrieval query (Sun et al., 2019; Azad & Deepak, 2019; Wang et al., 2020). Such coarse queries fail to capture a field's nuances and are never refined, leading to noisy and incomplete paper collections. **Second, the survey structure can be incoherent** (Fabbri et al., 2019; Gidiotis & Tsoumakas, 2020; Yang et al., 2023a). Since outlines are generated for each paper group independently and subsequently merged, the global structure lacks coherence and often misses important cross-group connections. **Third, injecting overly long contexts introduces distraction and context overload** (Liu et al., 2023; Wu et al., 2024). Feeding entire papers into LLMs not only

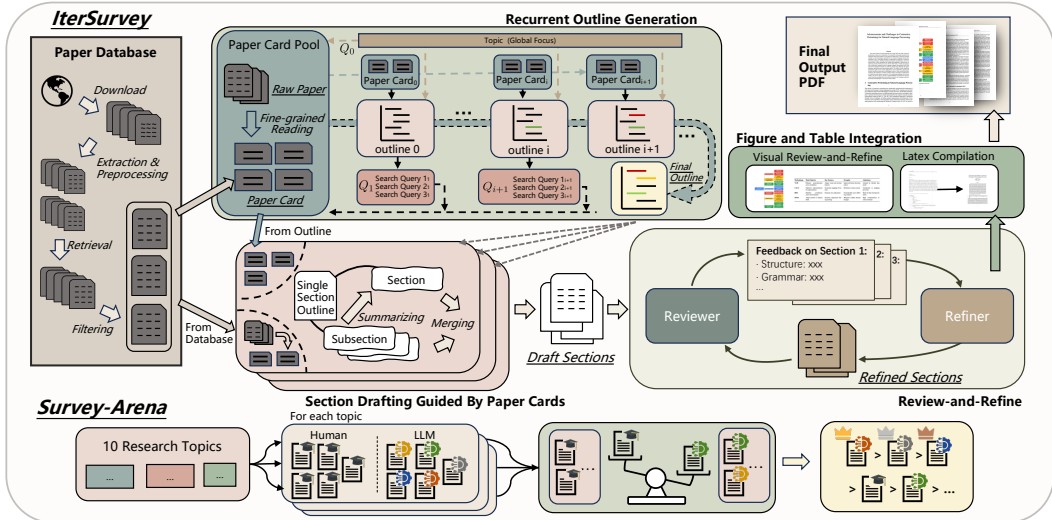

Figure 1: Overview of IterSurvey and Survey-Arena.

exposes them to large amounts of peripheral information, such as dataset details or experimental setups, which distracts from the conceptual structure needed for survey writing, but also places unnecessary pressure on the limited context window of the model.

In contrast, human researchers rarely attempt to grasp an entire field in a single shot. Instead, they follow an iterative reading process: starting with a small set of core papers, summarizing key contributions, and gradually expanding to related directions as their understanding deepens (Bates, 1989; Asai et al., 2023). Inspired by this workflow, we propose an iterative planning paradigm for automated survey generation, named as IterSurvey. IterSurvey shifts from a survey-centric pipeline to a paper-centric perspective, placing the understanding of individual papers at the core of the generation process. At its core lies a **recurrent outline generation** module that incrementally retrieves, organizes, and integrates evidence through a planning agent equipped with stability checks and stopping criteria, mitigating the brittleness of one-shot pipelines that rely on static queries and fragmented merges. Central to this process are **paper cards**, structured semantic abstractions that distill each paper into contributions, methods, and findings. Unlike conventional abstract-based inputs, these cards serve as fine-grained evidence units that guide both outline construction and **section drafting**, ensuring coherence and faithful citation across iterations. Finally, a **global review and integration** stage employs a reviewer–refiner loop to enforce consistency and clarity across sections, while an integrated figure–table generation pipeline compiles candidate visualizations, automatically checks them for layout and readability, and revises them to meet academic presentation standards. This design inherits the advantages of iterative human reading: retrieval is progressively refined rather than static (Jiang et al., 2023), the outline develops as an organically coherent structure rather than a patchwork (Zhang et al., 2025a), and paper cards enforce fine-grained evidence grounding that avoids distraction from peripheral details (Cachola et al., 2020; Wu et al., 2024).

Comprehensive experiments validate the effectiveness of our incremental paradigm. IterSurvey consistently outperforms all baselines across multiple dimensions, with recurrent outline generation yielding more coherent structures and paper cards improving citation accuracy without sacrificing precision. These advantages are further confirmed by human evaluation, where experts also favor the outputs of IterSurvey over competing systems. While these results confirm the superiority of IterSurvey, we find that absolute scoring struggles to reliably quantify the performance gap against human-written surveys (Yang et al., 2023b; Oren et al., 2023; Ye et al., 2024). In the LLM evaluation community, similar concerns have led to the development of Chatbot Arena Chiang et al. (2024), which adopts pairwise human preference judgments to overcome the noisiness and inconsistency of absolute ratings. Inspired by this paradigm, we further contribute **Survey-Arena**, the first benchmark to our knowledge that evaluates synthesized surveys through direct, pairwise ranking against a corpus of human-written exemplars. This approach provides a more robust and interpretable assessment of system quality by directly positioning it relative to a human-level baseline.

Our contributions are threefold.

- We propose **recurrent outline generation**, which iteratively retrieves, reads, and updates outlines with paper cards and outline–paper grounding, while encouraging the model to explore new directions.
- We develop a new framework: **IterSurvey**, which produces finer-grained outlines and supports multi-modal inputs and outputs for more comprehensive surveys.
- We construct **Survey-Arena**, a pairwise evaluation benchmark that ranks machine-generated surveys alongside human-written ones, enabling more reliable and interpretable assessment of survey quality.

## 2 RELATED WORK

**Automated Survey Generation**    Recent automated survey generation systems largely adopt a "one-shot" paradigm, where a static outline is constructed upfront before content generation. This approach is evident in pipeline-based systems like AutoSurvey (Wang et al., 2024b), which employs a hierarchical paradigm, and SurveyForge (Yan et al., 2025), which utilizes a memory-driven scholar navigation agent. Other frameworks focus on enhancing this initial outlining step through reference pre-processing; for instance, SurveyX (Liang et al., 2025) introduces an AttributeTree to extract key information, while HiReview (Hu et al., 2024) generates a hierarchical taxonomy tree. Tackling the challenge from a technical scalability perspective, SurveyGo (Wang et al., 2025) leverages the LLM×MapReduce-V2 algorithm to handle long contexts within this paradigm. In contrast, our framework treats the outline not as a static blueprint but as an evolving knowledge structure. Through a dynamic, recurrent mechanism, the outline is continuously updated as the system iteratively engages with the literature, resulting in comprehensive and coherent synthesis.

**Evaluation of Automated Surveys**    Evaluating machine-generated surveys is inherently challenging. Building on insights from automated peer review (Yu et al., 2024; Jin et al., 2024; Weng et al., 2025), prior works (Wang et al., 2024b; Yan et al., 2025; Liang et al., 2025) commonly adopt an LLM-as-a-judge paradigm with manually designed criteria, assessing dimensions such as coherence, coverage, and factuality. Citation quality is typically measured with NLI-based protocols (Gao et al., 2023), and Yan et al. (2025) additionally evaluate coverage by comparing system outputs with human-written surveys. While absolute scoring by LLMs provides useful fine-grained signals, it has also been noted to suffer from inconsistency and calibration issues (Ye et al., 2024; Latona et al., 2024), making system-level comparisons less reliable. In contrast, pairwise judgment which is widely used in chatbot evaluation (Zhao, 2025; Chiang et al., 2024) and peer review (Zhang et al., 2025b), offers more stable and interpretable assessments, but has not yet been applied to survey evaluation. To fill this gap, we introduce *Survey-Arena*, the first benchmark that ranks machine-generated surveys against human-written exemplars, providing both robust comparison across systems and a clearer positioning relative to human-level quality.

## 3 ITERSURVEY

An overview of IterSurvey is shown in Fig. 1, and its three core stages are detailed below.

### 3.1 RECURRENT OUTLINE GENERATION

Outline generation is a central component of automatic survey construction, as it requires understanding the research domain, identifying its subfields, and synthesizing individual papers. Alg. 1 shows the overview of the generation process. The outcome is a hierarchical framework that summarizes the domain, where each node in the hierarchy is represented by a title and an accompanying description. Given a topic query, our goal is to enable the model to integrate retrieval with inductive reasoning, so that it can systematically explore the literature and produce a comprehensive outline for the target domain. To this end, we design recurrent outline generation.

**Paper Card Pool.**    The paper card pool organizes retrieval keywords together with their associated papers in a structured mapping. For each keyword $K_i$, we retrieve $n$ candidate papers and extract $m$ of the most relevant references, forming the set:
$$\mathcal{P}_i = \{p_i^1, p_i^2, \ldots, p_i^{n+m}\}.$$

---

**Algorithm 1** Description of the recurrent outline generation process.

---

**Require:** Topic query $q$; retrieval sizes $(n, m)$; batch size $B$; paper budget $(N_{\min}, N_{\max})$; similarity threshold $\tau$

**Ensure:** Writing-oriented outline $\hat{O}$

1: $O \leftarrow$ INITOUTLINE($q$)
2: Pool $\leftarrow \emptyset$        ▷ map: query $\mapsto$ card list
3: $\mathcal{U} \leftarrow \emptyset$        ▷ consulted papers
4: $\mathbf{R} \leftarrow []$        ▷ query history
5: **for all** $r \in$ SEEDQUERIES($q$) **do**
6:     $\mathcal{P} \leftarrow$ RETRIEVE($r, n$) $\cup$ TOPREFS($\cdot, m$)
7:     $\mathcal{C} \leftarrow \{$PAPERCARD($p$) $\mid p \in \mathcal{P}\}$
8:     Pool[$r$] $\leftarrow \mathcal{C}$;    $\mathcal{U} \leftarrow \mathcal{U} \cup \mathcal{P}$
9: **while** $|\mathcal{U}| < N_{\max}$ **do**
10:     **if** Pool $= \emptyset$ **then**
11:        **if** $|\mathcal{U}| \geq N_{\min}$ **and** $h(O, \mathbf{R})$ **then**
12:           **break**
13:        **else**
14:           **for all** $r \in$ EXPANDQUERIES($O, \mathbf{R}$) **do**
15:              $\mathcal{P} \leftarrow$ RETRIEVE($r, n$) $\cup$ TOPREFS($\cdot, m$)
16:              $\mathcal{C} \leftarrow \{$PAPERCARD($p$) $\mid p \in \mathcal{P}\}$
17:              Pool[$r$] $\leftarrow \mathcal{C}$;    $\mathcal{U} \leftarrow \mathcal{U} \cup \mathcal{P}$
18:           **continue**
19:     $(r, \mathcal{C}) \leftarrow$ POP(Pool)        ▷ activate a query and its cards
20:     $\mathbf{R} \leftarrow \mathbf{R} \parallel r$
21:     **while** $\mathcal{C} \neq \emptyset$ **do**
22:        $\mathcal{B} \leftarrow$ SAMPLEBATCH($\mathcal{C}, B$)
23:        $\tilde{O} \leftarrow g(O, \mathcal{B}, r)$        ▷ retrieval + reading + synthesis
24:        **if** SIM($O, \tilde{O}$) $\geq \tau$ **then**
25:           $O \leftarrow \tilde{O}$
26:        $\mathcal{C} \leftarrow \mathcal{C} \setminus \mathcal{B}$
27: $\hat{O} \leftarrow$ REFINE($O$)
28: **return** $\hat{O}$

---

At iteration $i$, the system pops one keyword $K_i$ together with its associated paper set $\mathcal{P}_i$ from the pool. Each paper $p_i^j \in \mathcal{P}_i$ is converted into a paper card

$$c_i^j = \texttt{PaperCard}(p_i^j),$$

which distills the paper into its key information. In practice, a paper card is generated in a single structured pass following a manually designed schema that includes the problem motivation, core contributions, methodological summary, main findings, and limitations. This ensures that each paper card provides a consistent and comprehensive summary of the paper's essential elements. An example is shown in App. A.1. The collection of paper cards is denoted as $\mathcal{C}_i = \{c_i^1, c_i^2, \ldots, c_i^{|\mathcal{P}_i|}\}$. Overall, the paper card pool can be represented as a mapping

$$\mathcal{Q} = \{ K_i \mapsto \mathcal{C}_i \mid i = 0, 1, \ldots \},$$

where each keyword $K_i$ is associated with the corresponding set of paper cards $\mathcal{C}_i$.

**Outline updating.** The outline updating process begins with an empty initial outline, denoted as $O_0$. At each step, the outline is refined using the current outline $O_i$, the active keyword $K_i$, and a mini-batch of paper cards drawn from the pool. Specifically, let $\mathcal{B}_i \subseteq \mathcal{C}_i$ be a batch of paper cards sampled from the set of cards associated with $K_i$. The model produces a candidate update

$$\tilde{O}_{i+1} = g(O_i, \mathcal{B}_i, K_i),$$

where $g(\cdot)$ denotes the outline updating function. This procedure is repeated iteratively, with batches $\mathcal{B}_i$ of paper cards popped from the paper pool $\mathcal{Q}$ under the current keyword $K_i$, until all cards associated with $K_i$ are consumed and integrated into the outline. To ensure stability and promote refinement, the candidate update is accepted if its similarity to the previous outline exceeds $\tau$:

$$O_{i+1} = \begin{cases} \tilde{O}_{i+1}, & \text{if } \texttt{Sim}(O_i, \tilde{O}_{i+1}) \geq \tau, \\ O_i, & \text{otherwise.} \end{cases}$$

**Keyword expansion.** When all keywords $K_i$ has been fully consumed, the system explores new directions by proposing additional keywords. The goal is to identify potentially relevant aspects of the domain that have not yet been covered. Formally, new keywords are generated as

$$K_{i+1} = f(O_{i+1}, K_i, \ldots, K_0),$$

where $f(\cdot)$ denotes a keyword generation function that takes the updated outline and the history of queries as input, and proposes candidate keywords for further exploration. The corresponding paper set $\mathcal{P}_{i+1}$ is then retrieved and pushed into the pool $\mathcal{Q}$, thereby guiding the next iteration.

**Stopping condition.** Let $N_i = |\mathcal{P}_0 \cup \mathcal{P}_1 \cup \cdots \cup \mathcal{P}_i|$ denote the total number of consulted papers up to iteration $i$. The process terminates when either (i) $N_i \geq N_{\min}$ and the stopping signal

$$s = h(O_{i+1}, K_i, \ldots, K_0), \quad s \in \{0, 1\},$$

indicates that the outline is sufficiently complete, or (ii) $N_i \geq N_{\max}$. Here $h(\cdot)$ is a decision function which takes the evolving outline and the query history as input and outputs whether further exploration is necessary. This design ensures that the outline is not terminated prematurely, while also preventing excessive exploration.

**Post-processing.** After termination, the recurrent process produces a research-oriented outline $\tilde{O}$, which is further refined into a writing-oriented survey outline:

$$\hat{O} = \texttt{Refine}(\tilde{O}),$$

where $\texttt{Refine}(\cdot)$ reorganizes the structure, inserts standard survey components such as 'Introduction' and 'Future Directions', and ensures conformity with academic conventions. Finally, we perform paper–section relinking, where all consulted papers are reassociated with the corresponding sections of the final outline $\hat{O}$. This guarantees that each section of $\hat{O}$ is grounded in concrete evidence, providing a reliable foundation for subsection drafting.

### 3.2 SECTION DRAFTING GUIDED BY PAPER CARDS

A distinctive feature of our framework is that section drafting is entirely guided by paper cards, which serve as fine-grained, structured representations of the literature. Given the refined outline $\hat{O}$, each section or subsection is written by conditioning on its description $d_j$ together with the relevant pool of cards. Specifically, for a given subsection with description $d_j$, the system retrieves a set of additional reference papers $\mathcal{P}_{\text{sec}}^j$ and converts them into paper cards $\mathcal{C}_{\text{sec}}^j$. In contrast to previous work, our framework benefits from the paper–section relinking established during outline construction: each subsection is already associated with a pool of consulted papers from earlier iterations. This enriched evidence base, combining $\mathcal{C}_{\text{sec}}^j$ with the relinked cards, provides the model with a stronger foundation for subsection writing. Formally, the $j$-th subsection is generated as

$$S_j = \texttt{Draft}(d_j, \mathcal{C}_{\text{sec}}^j \cup \mathcal{C}_{\text{link}}^j),$$

where $\mathcal{C}_{\text{link}}^j$ denotes the set of paper cards relinked to subsection $j$. During drafting, the model is required to cite the provided references, and the citations are mapped to their corresponding papers.

### 3.3 GLOBAL REVIEW AND INTEGRATION

The final stage of survey generation goes beyond local drafting. It performs a global review-and-refine process that integrates sections into a coherent survey and enriches the survey with automatically generated figures and tables.

**Textual Review-and-Refine.** We adopt a reviewer–refiner loop that involves two collaborative LLM roles. The reviewer takes the entire survey draft as input to capture the global context but then focuses its critique on a specific section or subsection. This design ensures that feedback on local content is always grounded in an understanding of the overall narrative. The reviewer provides detailed suggestions covering aspects such as clarity of exposition, consistency of terminology, logical alignment with preceding and following sections, and stylistic fluency. The refiner then incorporates these suggestions to revise the targeted section, producing a polished update that fits better into the survey as a whole. This loop is applied sequentially across all sections and iterated multiple times, progressively enhancing readability, improving cross-section coherence, and strengthening the global structural integrity of the survey.

**Figure–Table Integration.**    In addition to textual refinement, we extend the refinement process to include multimodal elements, to further enhance readability.  For each section, the model first generates visualization requirements, such as tables with structured comparisons or figures with explanatory diagrams, together with natural language descriptions.  Based on these descriptions, candidate figures and tables are synthesized. The compiled outputs are then fed back to an LLM for quality assessment, enabling automatic detection of issues such as oversized layouts or unreadable text.  The LLM provides corrective suggestions, which are applied to improve the final visualizations. Finally, the text is refined again to ensure that all generated figures and tables are properly referenced within the survey.

# 4  EXPERIMENTS

## 4.1  EXPERIMENTAL SETTINGS

**Implementation Details.**    Following Wang et al. (2024b), we adopt **GPT-4o-mini** as our generation model for its balance of responsiveness and cost. Our retrieval database contains 680K computer science papers from arXiv, with PDFs converted into structured Markdown using MinerU (Wang et al., 2024a) for consistent formatting.  In outline generation, the system consults 1000–1200 papers, with a maximum of 8 sections. For section drafting, each subsection retrieves up to 60 additional relevant papers, combined with those linked during outline generation. Finally, we apply two iterations of the review-and-refine loop to enhance coherence across sections and improve overall readability. The details of the implementation are provided in App. A.2. Illustrative outputs compared with AutoSurvey are provided in App. A.13. A detailed analysis of the framework's time and cost overhead is provided in App. A.8.

**Baselines.**    We compare IterSurvey with a set of baselines, ranging from simple retrieval-augmented generation (Naive RAG), which directly drafts from retrieved documents, to more advanced state-of-the-art systems. Specifically, we evaluate against AutoSurvey (Wang et al., 2024b), the first systematic framework for this task; SurveyForge (Yan et al., 2025), which combines heuristic outline generation based on the logical structures of human-written surveys with a memory-driven scholar navigation agent for high-quality retrieval; and SurveyGo (Wang et al., 2025), which employs the LLM×MapReduce-V2 algorithm to address the long-context challenge. We also compare with SurveyX (Liang et al., 2025), which introduces an Attribute Tree-based outlining mechanism; however, due to access restrictions, we include SurveyX only in arena experiments. All methods are evaluated on the same retrieval database with generation hyperparameters aligned to their original settings for fairness.

## 4.2  AUTOMATIC EVALUATION RESULTS

**Evaluation Setup.**    We follow the evaluation protocol established in Wang et al. (2024b) and adopt their multi-dimensional LLM-as-a-judge framework. On the standard 20-topic suite used in Auto-Survey, we evaluate content quality along three dimensions: *coverage*, *structure*, and *relevance*, exactly following the criteria defined in Wang et al. (2024b). In addition, citation quality is assessed using the NLI-based evaluation of Gao et al. (2023), reporting both recall and precision: *Citation Recall* measures whether statements are fully supported by cited passages, while *Citation Precision* checks that citations are relevant and directly support the claims. To improve scoring stability and reliability, we standardize prompts and require judges to provide a rationale before assigning a score. For further robustness, we aggregate outputs from three independent LLM judges: GPT-4o, Claude-3.5-Haiku, and GLM-4.5V.[1] Full prompts are provided in App. A.12.

**Results.**    The results on the 20 topics from Wang et al. (2024b) are reported in Tab. 1. Statistical significance was confirmed via paired t-tests, indicating that IterSurvey consistently outperforms baseline models ($p < 0.05$). We summarize the main observations below.

- **Overall superiority.** IterSurvey consistently outperforms all baselines across both content and citation quality, achieving the highest overall average score (4.75). This demonstrates that the proposed framework is effective and robust across multiple evaluation dimensions.

---

[1]Specifically, we use `chatgpt-4o-latest`, `claude-3-5-haiku-20241022`, and `glm-4.5v`.

Table 1: Comparison of different methods in terms of content quality and citation quality.

| Methods | Content Quality | | | | Citation Quality | |
|---|---|---|---|---|---|---|
| | Coverage | Relevance | Structure | Avg. | Precision | Recall |
| NaiveRAG | $4.42_{\pm 0.50}$ | $4.85_{\pm 0.36}$ | $4.20_{\pm 0.73}$ | $4.49_{\pm 0.41}$ | $0.39_{\pm 0.16}$ | $0.40_{\pm 0.15}$ |
| AutoSurvey | $4.50_{\pm 0.29}$ | $4.80_{\pm 0.16}$ | $4.62_{\pm 0.24}$ | $4.64_{\pm 0.15}$ | $0.64_{\pm 0.08}$ | $0.64_{\pm 0.08}$ |
| SurveyForge | $4.57_{\pm 0.50}$ | $4.82_{\pm 0.39}$ | $4.60_{\pm 0.56}$ | $4.66_{\pm 0.40}$ | $0.59_{\pm 0.09}$ | $0.59_{\pm 0.09}$ |
| SurveyGo | $4.37_{\pm 0.49}$ | $4.83_{\pm 0.38}$ | $4.27_{\pm 0.63}$ | $4.49_{\pm 0.40}$ | $0.50_{\pm 0.11}$ | $0.63_{\pm 0.12}$ |
| IterSurvey | $\mathbf{4.58}_{\pm 0.50}$ | $\mathbf{4.95}_{\pm 0.22}$ | $\mathbf{4.72}_{\pm 0.45}$ | $\mathbf{4.75}_{\pm 0.30}$ | $\mathbf{0.64}_{\pm 0.06}$ | $\mathbf{0.70}_{\pm 0.07}$ |

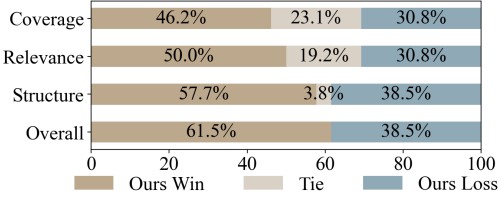

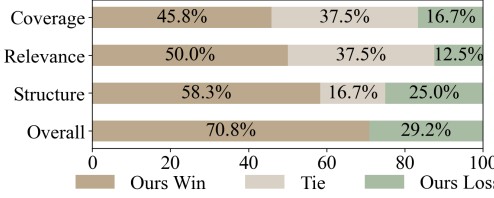

(a) IterSurvey vs AutoSurvey

(b) IterSurvey vs SurveyForge

Figure 2: LLM-generated survey comparison between AutoSurvey and IterSurvey.

- **Improved structural quality.** On the structure dimension, IterSurvey achieves the best score (4.72). This improvement stems from the recurrent outline generation mechanism, which iteratively explores the literature and refines the outline, resulting in clearer organizational planning and stronger cross-sectional coherence.

- **Enhanced citation quality.** IterSurvey also achieves superior citation performance. While maintaining the same precision as AutoSurvey, it improves recall to 0.70. This advantage is enabled by paper cards, which provide fine-grained summaries of individual papers and thus allow for retrieving and citing a broader yet still accurate set of supporting references.

Together, these results confirm that recurrent outline generation, paper cards, and outline–paper grounding synergize to produce surveys that are both structurally coherent and rigorously evidenced.

### 4.3 Human Evaluation Results

To further assess the quality of the generated surveys, we conducted a blind, pairwise study (Novikova et al., 2018; Chiang et al., 2024) with seven PhD-level experts. For each evaluation, experts were presented with an anonymized survey pair and asked to select the superior one based on multiple quality dimensions, including coverage, relevance, structural coherence, and overall quality, which is more objective and stable than ranking based on absolute scores (Herbrich et al., 2006; Sakaguchi et al., 2014). To control annotation cost, the human study was limited to direct comparisons between IterSurvey and two leading baselines: AutoSurvey and SurveyForge. Inter-rater agreement is reported in App. A.3. Results, as shown in Fig. 2, indicate that IterSurvey is consistently preferred over AutoSurvey and SurveyForge by domain experts, especially in terms of structure and overall quality. This trend aligns with our automatic evaluation, where recurrent outline generation also demonstrated stronger coherence and organization. The consistency between expert judgments and automatic metrics further highlights the robustness of IterSurvey in generating high-quality surveys.

### 4.4 Survey-Arena: Pairwise Comparison and Ranking

**Dataset construction.** Previous automatic evaluation methods typically assign an absolute score for each dimension, which struggles to fully capture the performance gap between machine-generated surveys and human-written ones. To move beyond absolute scores, we constructed the *Survey-Arena* benchmark. The benchmark spans ten research topics. For each topic, we manually selected five high-quality, human-written surveys to serve as a performance baseline. To ensure comparability, all surveys for a given topic were chosen from a narrow six-month submission window,

a process that required careful verification to ensure each topic had a sufficient number of suitable papers. We further confirmed their quality and influence via non-trivial citation counts on Google Scholar. The retrieval database for all machine-generated surveys was correspondingly frozen to the same time period to guarantee fairness. The full list of topics and papers is available in the App. A.7.

**Evaluation protocol.** For each topic, all possible pairs of a machine-generated survey and a human-written survey are constructed. To ensure robust evaluation and mitigate positional bias, each pair is judged in both directions (A vs. B and B vs. A), following Li et al. (2024). A panel of three distinct LLMs, namely GPT-4o, Claude-3.5-Haiku, and GLM-4.5V, serves as the judges for each comparison. Elo scores are computed from these aggregated pairwise outcomes to generate rankings for all systems.

**Results.** We report two key evaluation metrics: Avg. Rank, which indicates the mean position among all surveys, and >Human%, which reflects the proportion of topics where a system surpasses human surveys. The topic-wise outcomes from Survey-Arena are visualized in Fig.3, and the aggregated rankings are summarized in Tab.2.

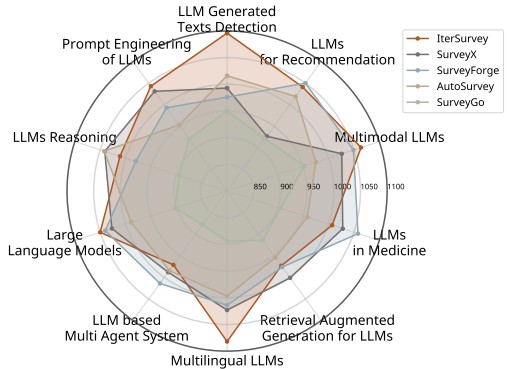

Each system is evaluated by its average rank across all surveys (including 5 machine-written surveys and 5 human-written ones) and by the proportion of topics where it surpasses human surveys. The results show that IterSurvey consistently achieves the best overall performance among automatic survey generation systems, with an average rank of 4.0 and surpassing human-written surveys in 60% of topics. These findings highlight that IterSurvey not only outperforms competing methods but also approaches human-level quality across diverse domains.

**Meta Evaluation.** To assess the reliability of Survey-Arena judgments, we compare the rankings produced by Survey-Arena for human-written surveys with citation counts on Google Scholar, which serve as an external signal of impact. Specifically, we compute Spearman's $\rho_s$ by measuring the correlation between Arena-derived and citation-based rankings for each topic, and then report the average across topics. For relevance scoring, we treat citation counts as an indicator of relevance and compute nDCG directly over the ranking lists. As a comparison, we also use the rankings derived from absolute scoring and compute their consistency and nDCG. This allows us to evaluate how well the different ranking methods align with citation-based rankings.

Results are shown in Tab. 3. Compared with the scoring-based approach, pairwise judgment achieves higher agreement with citation-based rankings, yielding a Spearman's $\rho_s$ of 0.410 and

Figure 3: Elo scores of Survey-Arena results across topics. The radar plot shows the Elo scores for each system across all topics, providing a topic-wise comparison.

Table 2: Aggregated rankings on Survey-Arena. Avg. Rank is the mean position among all surveys. >Human% is the average proportion of topics where a system surpasses human surveys.

| Method | Avg. Rank ↓ | > Human% ↑ |
|---|---|---|
| SurveyGo | 9.80 | 4% |
| AutoSurvey | 6.70 | 32% |
| SurveyForge | 4.80 | 50% |
| SurveyX | 4.70 | 54% |
| **IterSurvey** | **4.00** | **60%** |

Table 3: Consistency between different ranking methods and citation-based rankings.

| Rank Method | $\rho_s$ | nDCG@2 | nDCG@3 |
|---|---|---|---|
| Absolute Scoring | 0.320 | 0.695 | 0.767 |
| Pair-Judge | **0.410** | **0.834** | **0.873** |

nDCG@2/3 = 0.834/0.873. This indicates that when models are asked to directly compare two surveys, they more reliably identify the superior one, producing rankings that better align with human impact signals. These findings support pairwise evaluation as a more robust protocol for Survey-Arena.

Table 4: Comparison of different methods on survey-lacking topics.

| Methods | Content Quality | | | | Citation Quality | |
|---|---|---|---|---|---|---|
| | Coverage | Relevance | Structure | Avg. | Precision | Recall |
| AutoSurvey | $4.00_{\pm 1.12}$ | $4.20_{\pm 1.20}$ | $4.00_{\pm 1.00}$ | $4.07_{\pm 1.11}$ | $0.55_{\pm 0.14}$ | $0.55_{\pm 0.09}$ |
| SurveyForge | $\mathbf{4.50}_{\pm 0.50}$ | $4.75_{\pm 0.50}$ | $4.54_{\pm 0.54}$ | $4.60_{\pm 0.52}$ | $0.47_{\pm 0.12}$ | $0.47_{\pm 0.13}$ |
| IterSurvey | $4.42_{\pm 0.58}$ | $\mathbf{4.83}_{\pm 0.17}$ | $\mathbf{4.63}_{\pm 0.63}$ | $\mathbf{4.63}_{\pm 0.37}$ | $\mathbf{0.60}_{\pm 0.06}$ | $\mathbf{0.67}_{\pm 0.06}$ |

Table 5: Ablation study analyzing the contribution of each component in IterSurvey.

| Methods | Content Quality | | | | Citation Quality | |
|---|---|---|---|---|---|---|
| | Coverage | Relevance | Structure | Avg. | Precision | Recall |
| **Main** | $\mathbf{4.73}_{\pm 0.50}$ | $\mathbf{4.93}_{\pm 0.41}$ | $\mathbf{4.80}_{\pm 0.52}$ | $\mathbf{4.82}_{\pm 0.39}$ | $0.65_{\pm 0.04}$ | $\mathbf{0.77}_{\pm 0.04}$ |
| w/o Iterative Outline Paradigm | $4.53_{\pm 0.50}$ | $4.87_{\pm 0.31}$ | $4.53_{\pm 0.51}$ | $4.64_{\pm 0.41}$ | $\mathbf{0.66}_{\pm 0.08}$ | $0.70_{\pm 0.07}$ |
| w/o PaperCard | $4.52_{\pm 0.51}$ | $4.81_{\pm 0.38}$ | $4.52_{\pm 0.52}$ | $4.62_{\pm 0.39}$ | $0.63_{\pm 0.08}$ | $0.72_{\pm 0.06}$ |
| w/o Review/Refine | $4.60_{\pm 0.51}$ | $4.80_{\pm 0.42}$ | $4.60_{\pm 0.52}$ | $4.69_{\pm 0.39}$ | $0.64_{\pm 0.09}$ | $0.71_{\pm 0.08}$ |
| AutoSurvey | $4.53_{\pm 0.51}$ | $4.73_{\pm 0.43}$ | $4.47_{\pm 0.51}$ | $4.58_{\pm 0.44}$ | $0.57_{\pm 0.10}$ | $0.57_{\pm 0.09}$ |

## 4.5 Generalization on Survey-Lacking Topics

To examine whether automated survey generation can succeed in areas without existing surveys, we construct a subset of eight research topics (listed in App. A.9) where no human-written reviews are available. Such settings are common in emerging domains and pose greater challenges, since there are no canonical structures to imitate and the literature is often sparse and fragmented. This setup tests whether a system can autonomously organize the field into a coherent, well-grounded survey.

We compare IterSurvey against AutoSurvey and SurveyForge under this setup, and the results are presented in Tab. 4. Our method achieves the highest average score (4.63), consistently outperforming both baselines across content and citation quality. Notably, IterSurvey shows clear advantages in structural quality (4.63) and citation recall (0.67). Instead of fixed retrievals, our recurrent outline generation and paper card mechanism drive iterative exploration. This ensures structural coherence and broader reference coverage, even in domains where survey conventions are absent. Detailed results on these subsets are provided in App. A.9.

## 4.6 Ablation Study

We conduct an ablation study over five representative topics to quantify the contributions of the three core modules of IterSurvey: the Iterative Outline Paradigm, PaperCard, and the Review-and-Refine stage. Results are shown in Tab. 5, revealing the following insights:

**Iterative Outline Paradigm improves content organization.** Removing the iterative outline mechanism and replacing it with a one-shot outline generation results in clear degradation across all content-quality dimensions (Coverage: 4.53 vs. 4.73, Structure: 4.53 vs. 4.80). This demonstrates that iterative exploration helps the model achieve broader coverage and stronger organizational coherence by progressively integrating evidence. To further examine the effect of iterative planning itself, we additionally evaluate the quality of intermediate outlines produced at different stages of the pipeline, implementation details are shown in App. A.10. As shown in Fig. 5, outline quality increases consistently across iterations, rising from 3.67 to 4.46. Early rounds introduce most of the structural and technical improvements, while later rounds provide steady refinement. This confirms that iterative planning yields incremental gains throughout the generation process.

**PaperCard enhances citation grounding.** When replacing PaperCards with abstract-based inputs, citation recall drops from 0.77 to 0.72 while overall content quality also decreases. This indicates that structured paper-level distillation provides more complete and faithful evidence grounding, enabling the model to cite more comprehensively without sacrificing precision.

**Review-and-Refine provides additional polishing.** Omitting the review–and–refine stage reduces both content quality (Avg. from 4.82 to 4.69) and citation recall (0.77 to 0.71). These improvements show that iterative self-critique strengthens factual support, fills evidence gaps, and improves the overall coherence and readability of the final survey.

In addition, we extend our evaluation to assess the framework's robustness across different base models (App A.5) and its generalization capability in disciplines beyond Computer Science, such as Optimization (App A.6).

## 5 CONCLUSION

In this work, we tackled the limitations of existing survey generation systems by introducing Iter-Survey, a framework with recurrent outline generation, paper cards, and global review and integration. This design enables precise retrieval, coherent structure, and faithful citation grounding, while supporting multimodal outputs. Experiments on diverse topics show that IterSurvey outperforms state-of-the-art baselines in coherence, coverage, and citation quality. We also proposed Survey-Arena, a pairwise benchmark that complements absolute scoring for a more reliable assessment. Future work will extend our framework to broader domains, integrate richer multimodal evidence, and refine evaluation protocols toward human-level quality.

ETHICAL CONSIDERATIONS

Our work focuses on automatic literature survey generation using large language models. While the system is designed to support researchers by synthesizing existing knowledge, it inevitably inherits limitations of current models, including potential citation errors, incomplete coverage, and occasional inaccuracies. Therefore, the generated surveys are intended as an assistive tool rather than a substitute for human scholarship, and should be used for reference only. For evaluation, all human experts involved in the study participated voluntarily and received fair compensation. All data used in our experiments were sourced from publicly available arXiv papers, which permit non-commercial use. We strictly avoided the use of private or sensitive data.

USE OF LARGE LANGUAGE MODELS

We used large language models (GPT-4o, Claude-3.5-Haiku, and GLM-4.5V) in two ways: (i) as evaluation judges for assessing survey quality, and (ii) for limited language editing and refinement of the manuscript. All substantive research ideas, experimental design, analyses, and final decisions were made solely by the authors, who take full responsibility for the content of this paper.

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

# A    APPENDIX

## A.1    PAPER CARD EXAMPLE

Paper Card Data Structure Example

```
{
  "title": "Attention Is All You Need",
  "paper_type": "Research",
  "motivation_problem": "Traditional sequence transduction models
    (RNNs, CNNs) process data sequentially, which precludes
    parallelization and increases training time. They also
    struggle with learning dependencies between distant positions
    in a sequence.",
  "method_contribution": "The paper proposes the Transformer, a
    novel model architecture eschewing recurrence and relying
    entirely on an attention mechanism to draw global
    dependencies between input and output. Key components include
    Multi-Head Attention and Positional Encoding.",
  "results_findings": "On the WMT 2014 English-to-German
    translation task, the Transformer achieves 28.4 BLEU,
    improving over the existing best results, including
    ensembles, by over 2 BLEU. It also trains significantly
    faster than architectures based on recurrent or convolutional
    layers.",
  "limitations_future_work": "The quadratic complexity of
    self-attention with respect to sequence length limits the
    model's application on very long sequences. Future work
    includes extending the Transformer to input and output
    modalities other than text.",
  "related_papers": [
    "Long Short-Term Memory",
    "Convolutional Sequence to Sequence Learning",
    "Layer Normalization",
    "Neural Machine Translation by Jointly Learning to Align and
    Translate"
  ],
  "relevance_score": 5
}
```

## A.2    IMPLEMENTATION DETAILS

**For the retrieval process**. We implemented a lightweight database to provide the necessary functionality. The retrieval logic is based on vector similarity, using the *nomic-ai/nomic-embed-text-v1.5* (Nussbaum et al., 2024) embedding model with all hyperparameters set to their default values. Given a query, the database computes the similarity between the query vector and all paper vectors, and returns the top-k most relevant entries. In addition, the database supports bidirectional lookup between a paper's arXiv identifier and title, as well as filtering papers published prior to a specified cutoff date.

**For Similarity Threshold $\tau$.** In our recurrent outline generation module, the iterative updates produced by the LLM are susceptible to instability, particularly when addressing broad topics or when the model over-prioritizes partial evidence from newly retrieved documents. To maintain structural coherence, we validate the updated outline $O^{(t)}$ against its predecessor $O^{(t-1)}$, accepting the update only if their similarity exceeds a threshold $\tau$. This gating mechanism prevents the outline from collapsing into degenerate states. The value of $\tau$ was determined empirically based on pilot experiments. We found that setting $\tau$ too low (e.g., $< 0.5$) permits over-aggressive restructuring, leading to truncated outputs, the deletion of significant sections, and structural invalidity, which collectively undermine the iterative process. Conversely, setting $\tau$ too high (e.g., $> 0.90$) renders the model

overly conservative, causing it to reject legitimate refinements and thereby diminishing the outline's exploratory capability.

### A.3 RESULTS OF INTER-RATER AGREEMENT

To assess the reliability of human annotations and the consistency between human and machine evaluations, we computed Cohen's kappa coefficients across four evaluation dimensions: Coverage, Relevance, Structure, and Overall, as shown in Tab. 6. These results show substantial agreement both among human annotators and between human and machine evaluations, supporting the reliability and consistency of the evaluation process.

Table 6: Cohen's kappa coefficient between LLMs and human evaluations.

| Evaluation Pair | Coverage | Relevance | Structure | Overall |
|---|---|---|---|---|
| Human vs. Mixture of LLMs | 0.726 | 0.562 | 0.590 | 0.615 |
| Human vs. Human | 0.714 | 0.583 | 0.611 | 0.650 |

### A.4 TOPICS FOR AUTOMATIC EVALUATION

We utilize 20 topics derived from AutoSurvey (Wang et al., 2024b). Each topic is paired with a human survey, as shown in Tab. 7, which also reports the survey titles, arXiv IDs, and their latest citation counts from Google Scholar.

Table 7: Topics for Automatic Evaluation

| Topic | Human Survey | ArXiv ID | Citations |
|---|---|---|---|
| In-context Learning | A Survey on In-context Learning | 2301.00234 | 2396 |
| LLMs for Recommendation | A Survey on Large Language Models for Recommendation | 2305.19860 | 596 |
| LLM-Generated Texts Detection | The Science of Detecting LLM-Generated Texts | 2310.14724 | 308 |
| Explainability for LLMs | Explainability for Large Language Models: A Survey | 2309.01029 | 875 |
| Evaluation of LLMs | A Survey on Evaluation of Large Language Models | 2307.03109 | 4020 |
| LLMs-based Agents | A Survey on Large Language Model based Autonomous Agents | 2308.11432 | 1906 |
| LLMs in Medicine | A Survey of Large Language Models in Medicine | 2311.05112 | 217 |
| Domain Specialization of LLMs | Domain Specialization as the Key to Make Large Language Models Disruptive | 2305.18703 | 217 |
| Challenges of LLMs in Education | Practical and Ethical Challenges of Large Language Models in Education | 2303.13379 | 722 |
| Alignment of LLMs | Aligning Large Language Models with Human: A Survey | 2307.12966 | 435 |
| ChatGPT | Harnessing the Power of LLMs in Practice: A Survey on ChatGPT and Beyond | 2304.13712 | 1254 |
| Instruction Tuning for LLMs | Instruction Tuning for Large Language Models: A Survey | 2308.10792 | 1174 |
| LLMs for Information Retrieval | Large Language Models for Information Retrieval: A Survey | 2308.07107 | 544 |
| Safety in LLMs | Towards Safer Generative Language Models | 2302.09270 | 13 |
| Chain of Thought | A Survey of Chain of Thought Reasoning: Advances, Frontiers and Future | 2309.15402 | 290 |
| Hallucination in LLMs | A Survey on Hallucination in Large Language Models | 2311.05232 | 2599 |
| Bias and Fairness in LLMs | Bias and Fairness in Large Language Models: A Survey | 2309.00770 | 1009 |
| Large Multi-Modal Language Models | Large-scale Multi-Modal Pre-trained Models: A Comprehensive Survey | 2302.10035 | 285 |
| Acceleration for LLMs | A Survey on Model Compression and Acceleration for Pretrained Language Models | 2202.07105 | 101 |
| LLMs for Software Engineering | Large Language Models for Software Engineering: A Systematic Literature Review | 2308.10620 | 1058 |

### A.5 EFFECT OF DIFFERENT BASE MODELS

To investigate the extent to which the underlying base model rigidly determines IterSurvey's performance, we conducted additional experiments using GPT-4o, GPT-4.1-mini, and the original GPT-4o-mini. Due to costs, we evaluated these on a randomly sampled subset of 5 topics. The results shown in Tab. 8 indicate that IterSurvey remains consistently effective across all tested models, demonstrating its robustness regardless of the base model. Furthermore, our method exhibits strong scalability: performance improves significantly as the capability of the base model increases (e.g., GPT-4o outperforms GPT-4o-mini). This suggests that while the base model sets a performance baseline, IterSurvey effectively leverages stronger reasoning capabilities to achieve superior results.

### A.6 RESULTS ON OPTIMIZATION DOMAIN

To evaluate the generalization capability of IterSurvey in disciplines beyond standard Computer Science, we conducted additional experiments on five representative topics within the Optimization domain, which is shown in Tab. 9.

Table 8: Performance comparison of IterSurvey using different base models (evaluated on a random subset of 5 topics).

| Base Model | Content Quality | | | | Citation Quality | |
|---|---|---|---|---|---|---|
| | Coverage | Relevance | Structure | Avg. | Precision | Recall |
| GPT-4o | $4.75_{\pm 0.37}$ | $4.94_{\pm 0.09}$ | $4.83_{\pm 0.38}$ | $4.84_{\pm 0.20}$ | $0.68_{\pm 0.03}$ | $0.76_{\pm 0.04}$ |
| GPT-4o-mini | $4.40_{\pm 0.51}$ | $4.73_{\pm 0.45}$ | $4.67_{\pm 0.52}$ | $4.60_{\pm 0.51}$ | $0.66_{\pm 0.02}$ | $0.77_{\pm 0.04}$ |
| GPT-4.1-mini | $4.58_{\pm 0.35}$ | $4.75_{\pm 0.51}$ | $4.67_{\pm 0.50}$ | $4.67_{\pm 0.42}$ | $0.74_{\pm 0.03}$ | $0.83_{\pm 0.04}$ |

Table 9: List of evaluated topics in the optimization domain.

| Category | Topic |
|---|---|
| **Optimization** | Stochastic Optimization for Large-Scale Learning |
| | Zeroth-Order Optimization Methods |
| | Combinatorial and Integer Optimization |
| | Distributed and Federated Optimization |
| | Multi-Objective Optimization and Pareto Methods |

The results (shown in Tab. 10) demonstrate that IterSurvey maintains robust performance in this adjacent domain, consistently outperforming baseline methods. Specifically, IterSurvey achieves the highest average content quality score of 4.73, surpassing both SurveyForge (4.62) and AutoSurvey (4.60). Notably, our framework exhibits a significant advantage in evidence grounding, achieving a citation precision of 0.70, which is substantially higher than AutoSurvey (0.61) and SurveyForge (0.57). This confirms that the iterative outline generation and paper-card mechanism can effectively synthesize high-quality surveys with accurate citations, even in mathematically intensive fields like Optimization.

Table 10: Performance comparison of different methods on five optimization domain topics.

| Methods | Content Quality | | | | Citation Quality |
|---|---|---|---|---|---|
| | **Coverage** | **Relevance** | **Structure** | **Avg.** | **Precision** |
| AutoSurvey | $4.53_{\pm0.52}$ | $4.80_{\pm0.41}$ | $4.47_{\pm0.74}$ | $4.60_{\pm0.51}$ | $0.61_{\pm0.08}$ |
| SurveyForge | $4.40_{\pm0.51}$ | $4.93_{\pm0.26}$ | $4.53_{\pm0.52}$ | $4.62_{\pm0.36}$ | $0.57_{\pm0.11}$ |
| IterSurvey | $4.60_{\pm0.51}$ | $4.93_{\pm0.26}$ | $4.67_{\pm0.49}$ | $4.73_{\pm0.32}$ | $0.70_{\pm0.06}$ |

## A.7 TOPICS FOR SURVEY-ARENA

To construct the Survey-Arena benchmark, we select 10 topics, with several derived from Auto-Survey (Wang et al., 2024b) and SurveyForge (Yan et al., 2025). For each topic, we include 5 human-written surveys, requiring that their arXiv submission dates fall within a six-month window. We report their latest Google Scholar citation counts as a measure of impact, as summarized in Tab. 11. For reproducibility, we also specify the exact arXiv version, since submission dates can vary considerably across different versions of the same paper.

Table 11: Topics for Survey-Arena

| Topic | Human Survey | ArXiv ID | Citations |
|---|---|---|---|
| Large Language Models | Large Language Models: A Survey | 2402.06196v3 | 1133 |
| | Large Language Models Meet NLP: A Survey | 2405.12819v1 | 86 |
| | History, Development, and Principles of Large Language Models-An Introductory Survey | 2402.06853v2 | 73 |
| | Recent Advances in Generative AI and Large Language Models | 2407.14962v1 | 68 |
| | Exploring the landscape of large language models: Foundations, techniques, and challenges | 2404.11973v1 | 5 |
| Multimodal LLMs | MM-LLMs: Recent Advances in MultiModal Large Language Models | 2401.13601v3 | 381 |
| | Multimodal Large Language Models: A Survey | 2311.13165v1 | 299 |
| | The Revolution of Multimodal Large Language Models: A Survey | 2402.12451v1 | 98 |
| | How to Bridge the Gap between Modalities: Survey on Multimodal Large Language Model | 2311.07594v1 | 43 |
| | A Review of Multi-Modal Large Language and Vision Models | 2404.01322v1 | 39 |
| Multilingual LLMs | Multilingual Large Language Model: A Survey of Resources, Taxonomy and Frontiers | 2404.04925v1 | 83 |
| | A Survey on Multilingual Large Language Models: Corpora, Alignment, and Bias | 2404.00929v2 | 55 |
| | A Survey on Large Language Models with Multilingualism | 2405.10936v1 | 40 |
| | Surveying the MLLM Landscape: A Meta-Review of Current Surveys | 2409.18991v1 | 12 |
| | Multilingual Large Language Models: A Systematic Survey | 2411.11072v2 | 9 |
| LLMs Reasoning | A Survey of Long Chain-of-Thought for Reasoning Large Language Models | 2503.09567v3 | 130 |
| | From System 1 to System 2: A Survey of Reasoning Large Language Models | 2502.17419v2 | 110 |
| | Advancing Reasoning in Large Language Models: Promising Methods and Approaches | 2502.03671v1 | 19 |
| | A Survey of Frontiers in LLM Reasoning | 2504.09037v1 | 17 |
| | Thinking Machines: A Survey of LLM based Reasoning Strategies | 2503.10814v1 | 9 |
| Prompt Engineering of LLMs | A Systematic Survey of Prompt Engineering in Large Language Models | 2402.07927v1 | 748 |
| | The Prompt Report: A Systematic Survey of Prompt Engineering Techniques | 2406.06608v2 | 182 |
| | Prompt Design and Engineering: Introduction and Advanced Methods | 2401.14423v4 | 117 |
| | A Survey of Prompt Engineering Methods in Large Language Models for Different NLP Tasks | 2407.12994v1 | 60 |
| | Efficient Prom pting Methods for Large Language Models: A Survey | 2404.01077v1 | 56 |
| Retrieval-Augmented Generation for LLMs | Retrieval-Augmented Generation for Large Language Models: A Survey | 2312.10997v5 | 2583 |
| | A Survey on RAG Meeting LLMs: Towards Retrieval-Augmented Large Language Models | 2405.06211v3 | 559 |
| | A Survey on Retrieval-Augmented Text Generation for Large Language Models | 2404.10981v2 | 119 |
| | Retrieval-Augmented Generation for Natural Language Processing: A Survey | 2407.13193v2 | 77 |
| | Retrieval Augmented Generation (RAG) and Beyond | 2409.14924v1 | 70 |
| LLM-based Multi-Agent System | A survey on large language model based autonomous agents | 2308.11432v7 | 1623 |
| | Multi-Agent Collaboration Mechanisms: A Survey of LLMs | 2501.06322v1 | 79 |
| | Large language model agent: a survey on methodology, applications and challenges | 2503.21460v1 | 19 |
| | Agentic large language models, a survey | 2503.23037v2 | 12 |
| | A Survey on LLM-based Multi-Agent System: | 2412.17481v2 | 3 |
| LLM-Generated Texts Detection | A Survey on LLM-Generated Text Detection: Necessity, Methods, and Future Directions | 2310.14724v2 | 210 |
| | A Survey on Detection of LLMs-Generated Content | 2310.15654v1 | 69 |
| | Towards Possibilities & Impossibilities of AI-generated Text Detection: A Survey | 2310.15264v1 | 46 |
| | Detecting chatgpt: A survey of the state of detecting chatgpt-generated text | 2309.07689v1 | 22 |
| | Decoding the AI Pen: Techniques and Challenges in Detecting AI-Generated Text | 2403.05750v1 | 13 |
| LLMs in Medicine | Large language models in healthcare and medical domain: A review | 2401.06775v2 | 246 |
| | A Survey on Medical Large Language Models | 2406.03712v1 | 53 |
| | A Comprehensive Survey of Large Language Models and Multimodal Large Language Models in Medicine | 2405.08603v1 | 46 |
| | Large Language Models for Medicine: A Survey | 2405.13055v1 | 37 |
| | A Comprehensive Survey on Evaluating Large Language Model Applications in the Medical Industry | 2404.15777v4 | 32 |
| LLMs for Recommendation | A Survey on Large Language Models for Recommendation | 2305.19860v4 | 508 |
| | Recommender Systems in the Era of Large Language Models (LLMs) | 2307.02046v2 | 479 |
| | A Comprehensive Survey of Language Modelling Paradigm Adaptations in Recommender Systems | 2302.03735v3 | 117 |
| | Large Language Models for Generative Recommendation: A Survey and Visionary Discussions | 2309.01157v1 | 116 |
| | How Can Recommender Systems Benefit from Large Language Models: A Survey | 2306.05817v4 | 104 |

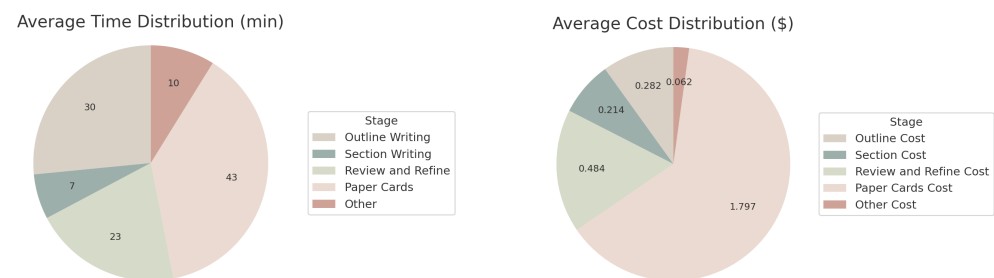

(a) Average time distribution across stages.   (b) Average cost distribution across stages.

Figure 4: Time and cost breakdown of LLM-generated survey pipeline.

### A.8  TIME AND COST ANALYSIS

To quantify the computational overhead of IterSurvey framework, we measure both time consumption and API cost across ten representative topics. We compute average time spent in each major stage of the workflow: outline writing, section writing, review-and-refine, paper card generation, and other operations (such as figure/table generation and LaTeX compilation) as shown in Fig. 4a. The corresponding API cost breakdown is presented in Fig. 4b. The reported time and cost are based on the use of GPT-4o-mini, and they may vary depending on the model and usage conditions. On average, generating a full survey requires approximately **113 minutes** and incurs a total API cost of **$2.84**, corresponding to a consumption of **16.2M input** and **661k output tokens**. Among all components, PaperCard generation accounts for the largest portion of both time and monetary cost. This is expected, as PaperCards require reading and distilling the full text of each retrieved paper, rather than relying solely on abstracts. While more expensive, this fine-grained evidence extraction substantially improves grounding quality. In practical deployment scenarios, however, this cost can be significantly reduced. PaperCards can be computed *offline* during corpus construction rather than at survey-generation time. Specifically, paper cards can be pre-generated using a lightweight LLM and stored in the retrieval database, with new papers being distilled immediately upon ingestion. This design amortizes PaperCard generation cost and avoids redundant recomputation, enabling fast and cost-efficient online survey generation.

### A.9  ANALYSIS OF SURVEY-LACKING SCENARIOS

We categorize the survey-lacking topics (Tab. 12) into two settings: *Survey-Absent* (unorganized literature) and *Literature-Sparse* (data-scarce). The detailed performance breakdown is provided in Tab. 13.

**Robustness in Survey-Absent Fields.** For topics like "*Event Timeline Generation*" where literature exists but lacks organization, IterSurvey achieves peak Coverage (4.53) and Citation Recall (0.67). This confirms that our *Paper Card* mechanism effectively synthesizes dispersed information, ensuring comprehensive coverage even without a structural template to follow.

**Coherence in Literature-Sparse Domains.** In domains such as "*RAG for Mechanical Design*," the supporting literature is relatively sparse. Even under such constraints, IterSurvey achieves strong Structural Quality (4.67), demonstrating that the *Recurrent Outline Generation* mechanism can construct coherent and well-organized outlines by leveraging broader domain knowledge when direct evidence is limited.

### A.10  ANALYSIS ON OUTLINE QUALITY ACROSS ITERATIONS

To evaluate how outline quality evolves during the recurrent outline generation process, we design an outline-structure criterion following the style of Wang et al. (2024b). The full rubric is shown below.

Table 12: Categorization of topics for Survey-Lacking Test.

| Category | Topic |
|---|---|
| **Survey-Absent** | Event Timeline Generation |
| | Agent-flow Data Curation |
| | Causal Mediation with Sparse Autoencoder Features in Transformers |
| | Multi-Tenant Scheduling for MoE Inference |
| | Renderer-in-the-Loop Supervision for Multimodal Model |
| | Linear RNN in Natural Language Processing |
| **Literature-Sparse** | Benchmarking Tool-Using LLMs for Causal Tasks in the MCP Ecosystem |
| | RAG for Mechanical Design: Cross-Modal Retrieval over CAD Trees and BOMs |

Table 13: Performance breakdown of IterSurvey on Survey-Absent and Literature-Sparse subsets.

| Settings | Content Quality | | | | Citation Quality | |
|---|---|---|---|---|---|---|
| | **Coverage** | **Relevance** | **Structure** | **Avg.** | **Precision** | **Recall** |
| Survey-Absent | $4.53_{\pm 0.45}$ | $4.84_{\pm 0.54}$ | $4.34_{\pm 0.45}$ | $4.57_{\pm 0.51}$ | $0.61_{\pm 0.03}$ | $0.67_{\pm 0.18}$ |
| Literature-Sparse | $4.33_{\pm 0.33}$ | $4.67_{\pm 0.33}$ | $4.67_{\pm 0.33}$ | $4.57_{\pm 0.26}$ | $0.57_{\pm 0.09}$ | $0.66_{\pm 0.03}$ |

---

**Outline Criterion**

```
Description: Outline quality is evaluated based on structural
    completeness and description depth. Sparse subsections,
    shallow single-sentence descriptions, and lack of named
    technical elements indicate insufficient depth, whereas rich
    subsection structure, detailed bullet points, and abundant
    technical terminology reflect strong depth.

Score 1: The outline is unusable, containing only keywords
    without coherent structure.
Score 2: Shallow outline with limited subsections and minimal
    technical specificity.
Score 3: Moderate outline with reasonable subsections and
    occasional technical mentions.
Score 4: Strong outline with well-developed subsections,
    structured lists, and frequent named methods.
Score 5: Exceptional outline with comprehensive structure,
    extensive bullet lists, and pervasive technical specificity.
```

Following the same evaluation setup as in our main experiments, three LLM judges independently score the outlines at different stages of the iterative process, and their averaged results are reported. As shown in Fig. 5, outline quality improves steadily from the initial to the final iteration (3.67 to 4.46), with early iterations contributing substantial structural expansion and later iterations providing consistent refinement. This validates that iterative planning brings incremental and meaningful improvements throughout the generation process.

## A.11 DETAIL OF NAIVE RAG

Given a topic, the Naive RAG system first retrieves 1,500 papers from the same database as ours. It then employs an iterative prompting strategy, where the LLM generates content until the total length of the survey reaches 5,000 tokens (Wang et al., 2024b). The prompt used for generation is shown below.

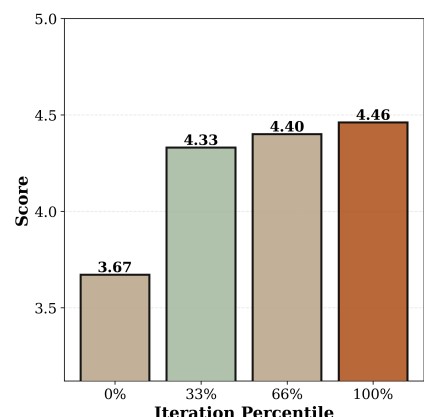

Figure 5: Outline quality across iterative refinement stages.

### Naive RAG Prompt

```
You are an expert in artificial intelligence who wants to write
    an overall and comprehensive survey about [TOPIC].

You are provided with a list of papers related to [TOPIC] below:
---
[PAPER LIST]
---

Here is the survey content you have written:
---
[SURVEY CONTENT]
---

Here is the requirement of the survey:
1. The survey must be more than [SURVEY LEN] tokens!
2. Containing several sections. Each section contains several
    subsections.
3. Cite several paper provided above to support the content you
    write.

Here is the format of your writing:
1. ## indicates the section title
2. ### indicates the subsection title
3. Only cite the "paper_title" in []. An example of citation: the
    emergence of large language models (LLMs) [Language models
    are few-shot learners; Language models are unsupervised
    multitask learners; PaLM: Scaling language modeling with
    pathways]

You need to continue writing the survey by adding a new section
    or subsection.

Do not stop until the length of survey is more than [SURVEY LEN]
    tokens!!!

Return the content you write:
```

## A.12 PROMPTS FOR EVALUATION

---

**NLI Prompt**

```
---
Claim:
[CLAIM]
---
Source:
[SOURCE]
---
Claim:
[CLAIM]
---
Is the Claim faithful to the Source?
A Claim is faithful to the Source if the core part in the Claim
    can be supported by the Source.\n
Only reply with 'Yes' or 'No':
```

---

**Criteria-based judging survey prompt**

```
You are an expert academic evaluator specializing in rigorous
    assessment of academic survey quality. Your task is to
    conduct a comprehensive evaluation using established
    scholarly standards and provide detailed justification for
    your assessment.

<topic>
[TOPIC]
</topic>

<survey_content>
[SURVEY]
</survey_content>

<instruction>
You are provided with:
1. A research topic for context
2. An academic survey for evaluation

Your task is to assess the survey quality based on the specific
    criterion provided below. Apply rigorous academic standards
    and provide detailed justification for your assessment. Base
    your evaluation on specific evidence from the survey content,
    considering both strengths and areas for improvement.
</instruction>

<evaluation_criterion>
Criterion Description: [Criterion Description]

**CRITICAL: Evaluation Standards**
Your evaluation must follow a systematic approach:

1. **Comprehensive Analysis**: Thoroughly examine the survey
    content against the specific criterion
2. **Evidence-Based Scoring**: Base your score on specific
    observable strengths and weaknesses
3. **Detailed Justification**: Provide specific examples and
    reasoning for your score
```

```
    **Scoring Framework**:
    Score 1: [Score 1 Description]
    Score 2: [Score 2 Description]
    Score 3: [Score 3 Description]
    Score 4: [Score 4 Description]
    Score 5: [Score 5 Description]

    </evaluation_criterion>

    <output_format>
    Provide your evaluation in the following structured format:

    **Rationale:**
    <Provide a comprehensive analysis of the survey's performance
        against the specific criterion. Include specific examples of
        strengths and weaknesses, with detailed justification for
        your assessment. Address how well the survey meets the
        criterion description and identify specific areas that align
        with or deviate from the scoring descriptions.>

    **Final Score:**
    <SCORE>X</SCORE>
    (Where X is the score from 1 to 5 based on your evaluation)

    Return your response in the following JSON format:
    {
      "rationale": "Your detailed reasoning here",
      "score": X
    }
    </output_format>

Now conduct your comprehensive evaluation of the academic survey
    quality.
```

## Coverage Criterion

```
Description: Coverage: Coverage assesses the extent to which the
    survey encapsulates all relevant aspects of the topic,
    ensuring comprehensive discussion on both central and
    peripheral topics.

Score 1: The survey has very limited coverage, only touching on a
    small portion of the topic and lacking discussion on key
    areas.
Score 2: The survey covers some parts of the topic but has
    noticeable omissions, with significant areas either
    underrepresented or missing.
Score 3: The survey is generally comprehensive in coverage but
    still misses a few key points that are not fully discussed.
Score 4: The survey covers most key areas of the topic
    comprehensively, with only very minor topics left out.
Score 5: The survey comprehensively covers all key and peripheral
    topics, providing detailed discussions and extensive
    information.
```

## Structure Criterion

```
Description: Structure: Structure evaluates the logical
    organization and coherence of sections and subsections,
    ensuring that they are logically connected.

Score 1: The survey lacks logic, with no clear connections
    between sections, making it difficult to understand the
    overall framework.
Score 2: The survey has weak logical flow with some content
    arranged in a disordered or unreasonable manner.
Score 3: The survey has a generally reasonable logical structure,
    with most content arranged orderly, though some links and
    transitions could be improved such as repeated subsections.
Score 4: The survey has good logical consistency, with content
    well arranged and natural transitions, only slightly rigid in
    a few parts.
Score 5: The survey is tightly structured and logically clear,
    with all sections and content arranged most reasonably, and
    transitions between adajecent sections smooth without
    redundancy.
```

## Relevance Criterion

```
Description: Relevance: Relevance measures how well the content
    of the survey aligns with the research topic and maintain a
    clear focus.

Score 1: The content is outdated or unrelated to the field it
    purports to review, offering no alignment with the topic.
Score 2: The survey is somewhat on topic but with several
    digressions; the core subject is evident but not consistently
    adhered to.
Score 3: The survey is generally on topic, despite a few
    unrelated details.
Score 4: The survey is mostly on topic and focused; the narrative
    has a consistent relevance to the core subject with
    infrequent digressions.
Score 5: The survey is exceptionally focused and entirely on
    topic; the article is tightly centered on the subject, with
    every piece of information contributing to a comprehensive
    understanding of the topic.
```

## Survey-Arena Review Prompt

```
# Paper 1:
Title: {title_1}
Figures: {figure_and_captions_1}
Content: {main_content_1}

# Paper 2:
Title: {title_2}
Figures: {figure_and_captions_2}
Content: {main_content_2}

You are provided with two survey papers on topic: {topic}.
```

```
As the area chair for a top ML conference, you can only select
    one paper. Start with a brief meta-review/reasoning of the
    pros and cons for each paper (two sentences), focusing on:

(1) insight and synthesis - moves beyond mere summarization to
    create new understanding and provides clear taxonomy;
(2) thoroughness and accuracy - comprehensive coverage of
    literature with technical correctness;
(3) structure and clarity - logical organization with compelling
    narrative;
(4) scope and impact - well-defined scope with valuable future
    research directions;
(5) presentation quality - professional polish, clear writing,
    and comprehensive evaluation of figures/tables presence and
    aesthetic quality.

Be very critical and do not be biased by what the author claimed.
    Finally, provide your choice in a binary format.

**Your Task:**
1. Provide a detailed evaluation for Paper 1 using the above
    criteria.
2. Provide a detailed evaluation for Paper 2 using the same
    criteria.
3. Make a final decision by comparing the two papers and
    justifying your choice.

STRICT OUTPUT INSTRUCTIONS:
- You MUST return a single valid JSON object.
- Output ONLY JSON. No explanations, no Markdown, no code fences,
    no additional text before or after the JSON.
- Use exactly these keys and types:
  - "paper_1_review": string
  - "paper_2_review": string
  - "chosen_paper": "1" or "2"
- Do NOT include any additional keys or trailing commas. If
    unsure, return empty strings for the review fields.

Return JSON in exactly this shape:
{
"paper_1_review": "Your meta-review and reasoning for paper 1",
"paper_2_review": "Your meta-review and reasoning for paper 2",
"chosen_paper": "1 or 2"
}

End your output immediately after the closing.
```

## A.13 COMPARISON BETWEEN AUTOSURVEY AND ITERSURVEY.

Figure 6: LLM-generated survey comparison between AutoSurvey and IterSurvey.

1. Introduction

datasets and generating actionable insights [12]. The ability of LLMs to produce sophisticated content, optimize marketing campaigns, and personalize user interactions marks a critical advancement in how businesses engage with customers and leverage AI technology.

In cybersecurity, LLMs are emerging as powerful tools to strengthen security measures and address vulnerabilities in digital infrastructures. Their adeptness in performing natural language processing tasks at scale equips them to analyze potential threats and detect anomalous patterns in vast datasets. Research highlights their potential to automate threat detection and incident response, enabling security professionals to respond more effectively to emerging threats [13]. Additionally, LLMs can generate insights from historical attack data, empowering organizations to preemptively mitigate risks and bolster their cybersecurity frameworks.

Beyond individual sectors, LLMs present opportunities for cross-industry solutions that tackle complex challenges. For example, in supply chain management, LLMs enhance predictive analytics by analyzing market trends and consumer behaviors, facilitating improved inventory management and logistics operations [14]. The integration of LLMs across various industries not only promotes operational efficiency but also fosters collaboration and innovation.

As LLMs continue to evolve, their role in enhancing accessibility to information becomes increasingly significant. They democratize expertise by granting individual access to sophisticated AI tools that were previously the domain of specialized professionals. This has critical implications for underserved communities, where LLMs can help bridge knowledge gaps in areas such as health education and legal assistance. By equipping users with pertinent information, LLMs empower them to make informed decisions that improve their quality of life [15].

The rapid development and deployment of LLMs underscore the importance of ethical considerations and the necessity for responsible AI use across all applications. As organizations adopt LLM technologies, addressing biases in training data, ensuring transparency of algorithms, and maintaining overall accountability in algorithmic decision-making become paramount. With potential risks arising from reliance on these models, it is crucial for stakeholders to establish guidelines and frameworks that ensure fairness, mitigate bias, and preserve user trust [16].

In conclusion, the significance of LLMs in modern AI is undeniable. Their transformative effects are evident across various sectors, with the capacity to revolutionize healthcare delivery, reshape educational practices, enhance business operations, strengthen cybersecurity efforts, and promote equitable access to information. Moving forward, ongoing research and collaboration among developers, researchers, and industry practitioners will be essential in realizing the full potential of LLMs while concurrently addressing the ethical challenges and societal implications of their increasing integration into our daily lives.

### 1.3 Human Interaction and LLMs

Large Language Models (LLMs) have emerged as a transformative force in the landscape of human-computer interaction (HCI), particularly in the realm of conversational AI. These models, capable of processing and generating human-like text, are fundamentally reshaping how users engage with technology across various domains. This subsection delves into the intricacies of human interaction with LLMs, emphasizing their facilitation of conversational AI, implications for user experience, and the factors that enhance user engagement.

Furthermore, effective alignment for LLMs requires an iterative process that embraces feedback from various stakeholders. Traditional training methods for AI systems often follow a linear approach, focusing primarily on model training and evaluation. However, the evolving nature of societal values and norms necessitates an adaptive approach to alignment that accommodates change over time. Stakeholders should be actively engaged in the ongoing assessment and refinement of AI systems to ensure they remain aligned with shifting human values. This perspective aligns with the notion of "Bidirectional Human-AI Alignment," wherein both AI systems and users are in a constant state of adaptation to each other [68].

Another significant sociotechnical challenge concerns the need for accountability and governance structures that can manage the complexities associated with AI deployment. As LLMs are increasingly integrated into decision-making processes across various domains—such as healthcare and criminal justice—the ramifications of misalignment become more pronounced. Establishing robust mechanisms for accountability and traceability in AI decision-making is essential. It is required AI systems to be designed with clear standards of transparency and governance so that stakeholders understand how alignment is achieved and can discuss the associated ethical considerations. Such frameworks help to mitigate risks related to misalignment and bolster public trust in AI systems [69].

The social context of AI deployment makes the implications of misalignment particularly poignant. For example, employing LLMs in high-stakes environments, such as education or law enforcement, necessitates balancing user autonomy with the potential for unintended and societal consequences. In these contexts, misalignment can lead to real-world repercussions, such as reinforcing existing inequalities or favoring certain groups over others. Clear guidelines, prioritizing ethical considerations and stakeholder engagement are instrumental in navigating these challenges and enhancing effective alignment strategies [70].

To promote successful alignment, organizations must also consider the broader impact of their AI systems on societal values. This involves recognizing not only the immediate outcomes of AI outputs but also how these systems can shape and influence public perception and behavior over time. As AI capabilities continue to evolve, the potential for consequential impacts grows. Acknowledging the interdependence of technology and society, researchers and practitioners must develop methodologies for assessing alignment that integrate sociotechnical perspectives to ensure enduring societal challenges [7].

In conclusion, adopting a sociotechnical perspective on the alignment of LLMs reveals the multifaceted nature of the challenges involved. By assuring collaborative approaches, ensuring diverse representation in AI development, integrating feedback via adaptive mechanisms, and establishing clear governance structures, stakeholders can work towards solving the effective alignment. Furthermore, by embedding ethical considerations and social implications into alignment strategies, the development of LLMs can better reflect the diverse values of society, ultimately enhancing trust and effectiveness in AI systems.

### 3 Techniques for Aligning LLMs

### 3.1 Reinforcement Learning from Human Feedback (RLHF)

Reinforcement Learning from Human Feedback (RLHF) represents a pivotal approach in aligning Large Language Models (LLMs) with human values and preferences. This framework enables models to learn desirable behaviors through direct interactions and feedback provided by human evaluators, rather than relying solely on traditional supervised training techniques. As LLMs are increasingly tasked with complex functions across diverse applications, RLHF has become essential for developing models that not only perform accurately but also align with user expectations in behavior and output.

methodologies. This survey aims to provide a thorough overview of the literature on LLM alignment, addressing the theoretical foundations and practical methodologies that have emerged in this rapidly evolving field [67]. Various alignment strategies, including Reinforcement Learning from Human Feedback (RLHF) [52] and Direct Preference Optimization (DPO) [37], are explored, alongside emerging frameworks for personalized and cultural alignment [50]. The structure of this survey is summarized in Figure 1, which outlines the key components and sections we will cover. By synthesizing insights from recent literature, this survey seeks to fill critical gaps in understanding how effective alignment can be achieved across diverse contexts and user demographics.

The historical context of alignment research reveals a progression from simplistic, rule-based systems to more sophisticated methods that consider the complexities of human values [98]. Early alignment strategies primarily focused on ensuring that AI systems adhered to predefined specifications, often overlooking the rich diversity of human preferences. Contemporary approaches leverage advanced techniques like RLHF, which utilizes human evaluations as reward signals, thereby allowing LLMs to refine their outputs based on user feedback [121]. However, RLHF presents challenges, such as high costs associated with gathering quality human feedback and the risk of bias stemming from limited or unrepresentative training data [17]. Additionally, while DPO offers a streamlined approach for optimizing model outputs based on user preferences, it too faces difficulties in capturing the multifaceted nature of human values across varied contexts [14]. The limitations inherent in these existing alignment frameworks underscore the necessity for more robust approaches that incorporate ethical considerations alongside user input in alignment processes, as recent studies advocate for the integration of fairness and accountability into these methodologies [26].

A pivotal aspect of alignment involves the integration of user feedback, enabling models to adapt to the dynamic and diverse experiences of users across various contexts [8]. Techniques such as active learning and adaptive learning strategies have been proposed to enhance alignment with individual user preferences, yet these methods often require considerable computational resources and may not scale effectively across varied populations [76]. Furthermore, the complexity of user preferences presents challenges in accurately capturing and utilizing feedback, necessitating robust methodologies that can translate user preferences into actionable insights for model training [6]. Research indicates that existing alignment methodologies frequently neglect the implications of biases, highlighting the importance of developing frameworks that represent the diverse values present in human societies [121]. As we explore alignment, it becomes evident that understanding these distinctions is crucial in addressing the challenges posed by the dynamic nature of human preferences and the limitations of current methodologies, which often rely on simplistic models that do not capture the nuances of real-world interactions.

In summary, this survey emphasizes that the alignment of LLMs with human preferences is an undertaking requiring in-depth engagement with the underlying theories, methodologies, and ethical considerations involved [33]. By fostering a deeper understanding of alignment as a multifaceted challenge, the research community can work towards creating AI systems that are not only technically proficient but also socially responsible and reflective of the values of the communities they serve [92]. The exploration of personalized alignment strategies is particularly vital as user interactions

3

3. Theoretical Foundations of Alignment

## 3 Theoretical Foundations of Alignment

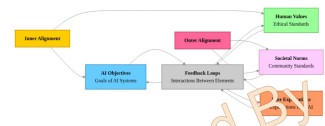

Figure 2: Conceptual Framework of Inner and Outer Alignment in AI Systems

The alignment of Large Language Models (LLMs) with human values is a pivotal area of research that underpins effective interactions between AI systems and users. This section will explore the theoretical foundations of alignment, including the distinctions between inner and outer alignment, the evolution of alignment methodologies, the challenges posed by biases, and the implications of cultural representation. Inner alignment focuses on the congruence of an AI's learned objectives with human values, while outer alignment pertains to how well AI behavior aligns with broader societal norms [62]. This dual framework reveals the multifaceted nature of alignment challenges, such as the dynamic variability of human values and contextual factors that shape user preferences, which can significantly differ across cultural landscapes [38, 75]. Understanding these distinctions is crucial as we examine the historical advancements in alignment methodologies, which have transitioned from rigid, rule-based systems to more flexible, data-driven approaches capable of adapting to evolving human expectations [77, 99]. Early AI systems relied heavily on fixed programming that proved inadequate for capturing the complexities of human behavior. In contrast, contemporary approaches leverage reinforcement learning techniques to enhance responsiveness to user input [46].

The conceptual framework of alignment, illustrated in Figure 2, highlights the interplay between inner and outer alignment in AI systems. However, defining and implementing effective alignment strategies is complicated by the limitations of existing frameworks, which often struggle to encapsulate the rich diversity of human experiences. Issues of bias and fairness have emerged as significant concerns, particularly as models trained on predominantly Western datasets may misrepresent the values of non-Western cultures, leading to misalignment in diverse applications [77, 99]. Research suggests that existing alignment methodologies frequently overlook the implications of these biases [62], highlighting the need for frameworks that are not only technically robust but also ethically sound and culturally aware. Approaches such as the Cultural Alignment Test (CAT) and the LLM-GLOBE benchmark have been introduced to systematically evaluate how well LLMs align with cultural values, revealing critical insights into areas of misalignment that require attention [54]. Furthermore, integrating user feedback mechanisms is essential; these systems must continuously adapt to individual user preferences, thereby necessitating a deeper understanding of how human values can inform model development and evaluation processes [85].

5

Figure 7: LLM-generated survey comparison between AutoSurvey and IterSurvey.

1404
1405
1406
1407
1408
1409
1410
1411
1412
1413
1414
1415
1416
1417
1418
1419
1420
1421
1422
1423
1424
1425
1426
1427
1428
1429
1430
1431
1432
1433
1434
1435
1436
1437
1438
1439
1440
1441
1442
1443
1444
1445
1446
1447
1448
1449
1450
1451
1452
1453
1454
1455
1456
1457

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

(a) AutoSurvey

REFERENCES

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

28

(b) IterSurvey

Figure 8: LLM-generated survey comparison between AutoSurvey and IterSurvey.

### 3.2 Strengths of ChatGPT

ChatGPT, developed by OpenAI, has garnered significant attention due to its impressive capabilities in generating coherent and contextually relevant responses across a variety of domains. A key strength of ChatGPT lies in its ability to engage in human-like conversation, effectively positioning it as a conversational agent across numerous applications. This conversational fluency is largely attributed to its underlying architecture, which employs the transformer model—an advancement that has substantially enhanced the effectiveness of many natural language processing (NLP) tasks [57].

The transformer architecture is particularly advantageous for text generation thanks to its self-attention mechanism. This feature enables the model to weigh the importance of different words in a sentence relative to one another, leading to a deeper understanding of context compared to previous models. As a result, ChatGPT generates responses that are not only grammatically correct but also contextually appropriate, demonstrating a higher level of coherence in its outputs [12].

Another notable strength of ChatGPT is its proficiency across a wide range of domains, which is facilitated by extensive training data encompassing various topics. This comprehensive training enables the model to produce information and answer queries on subjects spanning science, technology, arts, and personal advice. For example, studies indicate that ChatGPT can generate human-like responses in educational contexts, assisting students with inquiries across multiple disciplines [6]. By adapting its knowledge to the types of questions posed, ChatGPT delivers tailored responses, ultimately enhancing user engagement and satisfaction.

(a) AutoSurvey

### 5. Performance Evaluation of ChatGPT

Table 1: Performance Evaluation of ChatGPT across Different Domains. Abbreviations: MCQs = Multiple Choice Questions, USMLE = United States Medical Licensing Examination, F1 = F1 Score

| Domain | Metric | Accuracy | Strengths | Weaknesses |
|--------|--------|----------|-----------|------------|
| Education | MCQs | 56.9% | High accuracy in coding (73.2% on Leet-code) | Low accuracy in concepts (33.4% in DBMS) |
| Healthcare | USMLE | 58.2% (logical forming) | Potential utility in education | Lacks nuanced understanding (20% in Anatomy) |
| Legal | F1 Score | 0.49 (avg), ¿0.86 with guidance | Generates relevant content | Incomplete reasoning paths |

stakeholders remain vigilant in addressing the ethical and regulatory challenges they present, ensuring that the benefits of AI are realized without compromising ethical principles or consumer trust.

(b) IterSurvey

Figure 9: LLM-generated survey comparison between AutoSurvey and IterSurvey.

