# OpenReview forum: "Deep Literature Survey Automation with an Iterative Workflow"
_ICLR.cc/2026/Conference — Submitted to ICLR 2026_

### Official Review · Reviewer_11XG · 2025-10-23

**Soundness:** 2
**Presentation:** 2
**Contribution:** 2
**Rating:** 4
**Confidence:** 4

**Summary:**

This paper develops a new framework: IterSurvey, which produces finer-grained outlines and supports multi-modal inputs and outputs for more comprehensive surveys, and constructs Survey-Arena, a pairwise evaluation benchmark that ranks generated surveys alongside human-written ones, enabling more reliable and interpretable assessment of survey quality.

**Strengths:**

1. An iterative mechanism is designed for survey generation and optimization.

2. Integration that includes multi-modal elements.

3. Introduce the Survey-Arena pairing evaluation benchmark.

**Weaknesses:**

1. The methodology is not innovative enough, and the iterative mechanism or  review-and-refine are not novel contribution. What is the difference between it and autosurvey [1].

2. Does the introduction of iteration mechanism and paper card have a big impact on the time consumption of the whole framework?

3. The cost of different methods is not compared.

[1] Yidong Wang, Qi Guo, Wenjin Yao, Hongbo Zhang, Xin Zhang, Zhen Wu, Meishan Zhang, Xinyu Dai, Qingsong Wen, Wei Ye, et al. Autosurvey: Large language models can automatically write surveys. Advances in neural information processing systems, 37:115119–115145, 2024.

**Questions:**

1. What is the difference of he iterative mechanism and review-and-refine between IterSurvey and Autosurvey [1].
2. What are the impact of iteration mechanism and paper card on the time consumption of the whole framework?
3. The cost of different methods should be compared.
4. Did you replicate the experiment independently, generating only one review per topic for comparison?

[1] Yidong Wang, Qi Guo, Wenjin Yao, Hongbo Zhang, Xin Zhang, Zhen Wu, Meishan Zhang, Xinyu Dai, Qingsong Wen, Wei Ye, et al. Autosurvey: Large language models can automatically write surveys. Advances in neural information processing systems, 37:115119–115145, 2024.

---

> ### Author Response · Authors · 2025-11-24
> **Response to Reviewer 11XG (Part 1)**
>
> We thank the reviewer for the constructive feedback and for highlighting the strengths of our iterative workflow and multimodal integration. The reviewer raises several important points regarding methodological novelty, computational efficiency, and evaluation fairness. We address each concern point by point below, and all corresponding analyses and experiments have been incorporated into the revised manuscript.
>
> ---
>
> > W1 & Q1: Novelty and difference from AutoSurvey [1]
> We appreciate the opportunity to clarify this point. IterSurvey is different from AutoSurvey in its design goals, workflow, and mechanisms for grounding and refinement. The key distinctions are:
>
> 1. **Iterative Outline Generation vs. One-Shot Outline (AutoSurvey)**
>   - AutoSurvey generates the entire outline in a single step.
>   - IterSurvey performs recurrent outline generation, where the outline is updated across multiple rounds with newly retrieved evidence. Each iteration integrates more papers, gradually expands coverage, and prevents early hallucinations. As shown in our ablation study, removing iterative outline generation reduces Structure by −0.33 and Coverage by −0.21.
>   - Importantly, this iterative process is what enables IterSurvey to excel on survey-lacking or emerging topics, where no human-written survey or stable structure exists. By continuously revising the outline based on fresh evidence, IterSurvey can construct a high-quality domain structure from scratch, rather than relying on templates or prior surveys. This capability is reflected in our out-of-distribution evaluation, where IterSurvey achieves strong Structure and Coverage scores even when only sparse or no survey literature exists.
>
> 2. **PaperCard Grounding vs. Tuncated Evidence (AutoSurvey)**
>   - AutoSurvey uses truncated main content or abstract of papers due to the context length limitation.
>   - IterSurvey introduces PaperCards, a schema-guided distillation including motivation, contributions, methodology, findings, and limitations.
>   - This enables finer-grained grounding, improving citation recall by +0.05 over abstract-based inputs and yielding more stable evidence integration.
>
> 3. **Multi-round Review-and-Refine vs. Single-Pass Improvement**
>   - AutoSurvey applies one coarse refinement pass.
>   - IterSurvey implements multiple review–refine rounds, each explicitly checking for missing evidence, unsupported claims, inconsistent structure, and weak transitions.
>   - This leads to improvements in both content quality and citation fidelity, as confirmed by ablations.
>
> These differences go beyond incremental modifications; they represent **a fundamental shift** in how surveys are generated. Whereas AutoSurvey constructs surveys based on pre-existing outlines and abstracts, IterSurvey takes a **paper-centric** approach, focusing on each individual paper’s contributions and using a larger context to **build a deep and coherent understanding of the domain**. This approach allows IterSurvey to better mimic how human researchers build knowledge of a field by reading papers, extracting core information, and iteratively refining their understanding.

---

> > ### Author Response · Authors · 2025-11-24
> > **Response to Reviewer 11XG (Part 2)**
> >
> > > W2 & Q2: Computational overhead and time cost
> >
> > We agree that the iterative workflow requires more computation. Our goal, however, is to improve survey quality, evidence grounding, and stability, rather than speed.
> >
> > To increase transparency, we added a detailed cost analysis in the appendix, based on experiments using GPT-4o-mini and averaged over ten topics.
> >
> > | Topic   | Outline Writing | Section Writing | Review and Refine | Paper Cards     | Other          | Total Time(min)     |
> > |---------|------------------|------------------|--------------------|------------------|----------------|----------------|
> > | AVG | 30      | 7      | 23        | 43      | 10    | 113    |
> >
> > | Topic   | Outline Cost | Section Cost | Review and Refine Cost | Paper Cards Cost | Other Cost | Total Cost |
> > |---------|--------------|--------------|--------------------------|-------------------|------------|------------|
> > | AVG | $0.282133    | $0.214191    | $0.484496                | $1.797310         | $0.061535  | $2.839665  |
> >
> > As shown, most overhead comes from PaperCard construction, which performs full-paper distillation rather than abstract-level summarization. While this step is more expensive, our ablation results confirm it is critical for reliable grounding.
> >
> > Importantly, PaperCards need not be generated online. In practical deployment, PaperCards can be precomputed offline and stored in the retrieval backend, making the cost largely amortized and eliminating repeated computation.
> >
> > > Q4: Evaluation protocol
> >
> > Yes. All systems (IterSurvey, AutoSurvey, SurveyForge, SurveyX, SurveyGo) generate one full survey per topic, following the standard AutoSurvey evaluation protocol[1] to ensure a fair, one-to-one comparison.
> > Each survey is then assessed by three independent LLM judges, who provide reasoning and scores across multiple dimensions (coverage, relevance, structure, citation precision/recall). Final results are averaged over topics and judges.
> >
> >
> > **Reference:**
> >
> > [1] Wang, Yidong, et al. "Autosurvey: Large language models can automatically write surveys." Advances in neural information processing systems 37 (2024): 115119-115145.

---

### Official Review · Reviewer_MrcU · 2025-10-29

**Soundness:** 2
**Presentation:** 3
**Contribution:** 2
**Rating:** 6
**Confidence:** 3

**Summary:**

This paper proposes an auto-literature review generation framework *i.e.*, IterSurvey. The core idea of IterSurvey is to mimic the iterative research process of human researchers, through a cyclic outline generation module, incrementally conducting literature retrieval, reading, and updating to ensure the exploratory and coherent nature of the review. Additionally, to more reliably evaluate the quality of the generated reviews, the paper constructs a paired evaluation benchmark named Survey-Arena, which provides a more robust performance ranking by directly comparing machine-generated reviews with those written by humans.

**Strengths:**

The paper abandons the static "one-time" planning mode and instead adopts a dynamic, iterative workflow which may iteratively improve the quality of the survey. Besides, injecting multimodal items (table and figure) into a survey is interesting.

**Weaknesses:**

1. Lack of efficiency and cost analysis: The iterative framework proposed in the paper, especially the cyclic generation of outlines and multi-round "review and optimization," seems to require a large amount of computational resources and time. However, the paper does not provide a comparative analysis of the efficiency (such as the time required to generate a review) and cost (API usage) between IterSurvey and other baseline methods.

2. The generation details of the "paper card" need to be supplemented: The process of constructing paper cards seems to need lots of time (using LLM to generate motivation, contribution, and so on). The authors need to add more details.

3. The authors need to verify the effectiveness of the iterative pipeline. For example, the performance improvement is seen with the increase in iterations.

**Questions:**

Please refer to the weakness.

---

> ### Author Response · Authors · 2025-11-24
> **Response to Reviewer MrcU (Part 1)**
>
> We thank the reviewer for the constructive feedback and for recognizing the value of our iterative workflow and multimodal integration. Below we address the reviewer’s concerns point by point.
>
> ---
>
> > W1: Lack of efficiency and cost analysis: The iterative framework proposed in the paper, especially the cyclic generation of outlines and multi-round "review and optimization," seems to require a large amount of computational resources and time. However, the paper does not provide a comparative analysis of the efficiency (such as the time required to generate a review) and cost (API usage) between IterSurvey and other baseline methods.
>
> The main contribution of IterSurvey is not to accelerate survey generation but to substantially improve survey quality through a human-like, evidence-driven workflow. The recurrent outline generation and multi-round review steps are designed to enhance topic coverage, grounding fidelity, and cross-sectional coherence. These capabilities are difficult to achieve with one-shot systems and naturally require additional computation.
>
> To increase transparency, we have added a detailed runtime and token-cost analysis in the revised version. The majority of overhead comes from PaperCard construction, which performs fine-grained paper-level distillation. Although more expensive, this step is directly responsible for the significant improvement in citation grounding observed in our ablations.
>
> The average cost per topic on gpt-4o-mini is summarized below:
>
> | Topic   | Outline Writing | Section Writing | Review and Refine | Paper Cards     | Other          | Total Time     |
> |---------|------------------|------------------|--------------------|------------------|----------------|----------------|
> | AVG | 30      | 7      | 23        | 43      | 10    | 113    |
>
> | Topic   | Outline Cost | Section Cost | Review and Refine Cost | Paper Cards Cost | Other Cost | Total Cost |
> |---------|--------------|--------------|--------------------------|-------------------|------------|------------|
> | AVG | $0.282133    | $0.214191    | $0.484496                | $1.797310         | $0.061535  | $2.839665  |
>
> In practical deployment, this cost can be substantially reduced because PaperCards can be precomputed and stored in the retrieval database, eliminating repeated online generation. Thus, the workflow remains practical while preserving its core quality advantages.
>
> > W2: The generation details of the "paper card" need to be supplemented: The process of constructing paper cards seems to need lots of time (using LLM to generate motivation, contribution, and so on). The authors need to add more details.
>
> Thank you for pointing out the need for more details about PaperCard generation. In the original submission, this part may not have been sufficiently explained. We have expanded it in the revised version.
>
> PaperCards are generated in a single structured pass, where the LLM is prompted to fill a manually designed schema that includes fields such as: motivation/problem, key methodological contributions, main findings, limitations, related-works, and a short list of referenced papers. This schema ensures that each card captures the essential aspects of a paper in a consistent and comprehensive format. We have added details to the updated manuscript. In the revised manuscript, we have added detailed examples and definitions in the appendix.

---

> > ### Author Response · Authors · 2025-11-24
> > **Response to Reviewer MrcU (Part 2)**
> >
> > > W3: The authors need to verify the effectiveness of the iterative pipeline. For example, the performance improvement is seen with the increase in iterations.
> >
> >
> > Thank you for suggesting a verification of the effectiveness of the iterative pipeline. We have conducted an additional analysis to evaluate how survey quality changes across iterations.
> >
> > To evaluate intermediate outlines, we designed an outline-quality criterion following the style of AutoSurvey’s scoring rubric. The rubric assesses structural completeness and description depth on a 1–5 scale. The full rubric is included in the revised manuscript.
> >
> > Using this rubric, we evaluated outlines generated at different stages of the iterative workflow (0 percent, 33 percent, 66 percent, and 100 percent of updates). Three LLM judges were used, and the reported numbers are averaged over the five ablation topics. The results are:
> >
> > | Metric        | Score |
> > |---------------|--------|
> > | 0% Iteration  | 3.67   |
> > | 33% Iteration | 4.33   |
> > | 66% Iteration | 4.40   |
> > | 100% Iteration| 4.46   |
> >
> > The trend shows monotonic improvement as iterations proceed. Early iterations contribute the largest gains by enriching subsection structure and introducing domain-specific technical elements. Subsequent iterations bring smaller but consistent refinements, improving organizational stability and aligning content more tightly with retrieved evidence.
> >
> > These findings provide direct empirical validation that iterative refinement is not merely cosmetic but substantively enhances both structure and coherence. The analysis also confirms that each iteration contributes meaningfully, supporting the design choice of a recurrent rather than one-shot planning mechanism. The full study and rubric details have been included in the revised version.
> >
> > ---
> >
> > We hope the above responses satisfactorily address your concerns and assist in your reassessment of our work. Please feel free to let us know if any further clarification is needed.

---

### Official Review · Reviewer_z89M · 2025-10-31

**Soundness:** 3
**Presentation:** 3
**Contribution:** 2
**Rating:** 4
**Confidence:** 4

**Summary:**

This paper proposes IterSurvey to fix flaws of one-shot automated survey systems (noisy retrieval, fragmented structure). It uses recurrent outline generation, paper cards (distilling papers’ key info), and a review loop. Experiments show it outperforms baselines (e.g., AutoSurvey) in coverage, coherence and citations. The authors also introduce Survey-Arena for reliable pairwise assessment against human surveys, noting IterSurvey is an assistive tool with data from public arXiv papers.

**Strengths:**

- The integration of paper cards and a review-and-refine loop with visualization enhancements significantly improves textual flow, cross-sectional coherence, and multimodal integration, as evidenced by the experimental results.

- The proposed Survey-Arena, a pairwise evaluation benchmark that compares machine-generated surveys with human-written ones, offers a more reliable assessment and effectively addresses the limitations of absolute scoring methods.

**Weaknesses:**

My main concerns are the limitations of the paper's coverage and the objectivity of the evaluation. If the author can address my questions, I would be happy to increase the rating:
- The retrieval database includes only 680K computer science papers from arXiv, resulting in limited coverage of other disciplines and constraining the framework’s generalization ability to non-CS domains.

- The iterative workflow—involving recurrent outline generation and multiple review-and-refine loops—requires repeated retrieval and model invocations, potentially leading to higher computational costs compared to one-shot systems.

- The human evaluation involves only seven PhD-level experts and compares performance against two baselines (AutoSurvey, SurveyForge), with a small sample size that may restrict the representativeness and robustness of the findings.

- The framework lacks evaluation on established benchmarks such as SurveyBench (proposed in SurveyForge), which would enhance the comparability and credibility of its performance assessment.

**Questions:**

- In the recurrent outline generation module of IterSurvey, how is the similarity threshold τ—which determines whether to accept candidate outline updates—specifically calibrated, and what effects do different values of τ have on the stability of the outline and its exploratory capability?

- Regarding paper cards, which distill papers into their core contributions, methods, and findings, does the framework adopt a specific criterion for deciding which details to retain or discard during the distillation process, and how does it prevent the omission of critical information?

- The ablation analysis presented in Table 5 appears insufficient; it would be preferable to conduct a single-component ablation study to more clearly assess the contribution of each module.

- In the comparative evaluation with baselines (e.g., AutoSurvey, SurveyForge), why was the analysis limited to the “content quality” and “citation quality” dimensions, without including a comparison of computational efficiency (e.g., time consumption and resource utilization)?

- While IterSurvey performs effectively on emerging topics lacking existing surveys, does it incorporate adaptation strategies for domains with extremely sparse literature, such as newly developed interdisciplinary fields?

- Finally, to what extent does the underlying base model rigidly determine the overall performance of IterSurvey?

---

> ### Author Response · Authors · 2025-11-24
> **Response to Reviewer z89M (Part 1)**
>
> We thank the reviewer z89M for the constructive and detailed feedback. The reviewer’s comments highlight important issues regarding coverage, evaluation objectivity, computational efficiency, and methodological clarity. We appreciate these insights, and we have conducted new analyses and additional experiments to address these points directly. All updated results have been incorporated into the revised paper. Below, we provide point-by-point responses and clarify the design choices:
>
> ---
>
> > W1: The retrieval database includes only 680K computer science papers from arXiv, resulting in limited coverage of other disciplines and constraining the framework’s generalization ability to non-CS domains.
>
> Thank you for raising this important point on generalization beyond the CS domain.  To directly address this concern, we conducted a new cross-domain experiment in the Optimization (opt) category of arXiv. Unlike Computer Science, Optimization is rooted in Operations Research and reflects a different research community, writing conventions, and conceptual structure.
>
> We selected five representative optimization topics:
>
> 1. Stochastic Optimization for Large-Scale Learning
> 2. Zeroth-Order Optimization Methods
> 3. Combinatorial and Integer Optimization
> 4. Distributed and Federated Optimization
> 5. Multi-Objective Optimization and Pareto Methods
>
> IterSurvey and two baselines (AutoSurvey, SurveyForge) were evaluated under the same LLM-as-a-judge protocol used in the main paper. The results are:
>
> | Method | Coverage | Relevance | Structure | Avg. | Precision | Recall |
> | :--- | :---: | :---: | :---: | :---: | :---: | :---: |
> | AutoSurvey | 4.53 ± 0.52 | 4.80 ± 0.41 | 4.47 ± 0.74 | 4.60 ± 0.51 | 0.61 ± 0.08 | 0.61 ± 0.08 |
> | SurveyForge | 4.40 ± 0.51 | 4.93 ± 0.26 | 4.53 ± 0.52 | 4.62 ± 0.36 | 0.57 ± 0.11 | 0.57 ± 0.11 |
> | **IterSurvey** | **4.60 ± 0.51** | **4.93 ± 0.26** | **4.67 ± 0.49** | **4.73 ± 0.32** | **0.70 ± 0.06** | **0.76 ± 0.04** |
>
> IterSurvey outperforms both baselines across all metrics, improving average content quality and achieving notable gains in citation precision and recall. These results indicate that the iterative workflow generalizes well beyond computer science, even in domains with substantially different writing structures and terminology. The full cross-domain evaluation has been added to the revised paper.
>
> > W2: The iterative workflow—involving recurrent outline generation and multiple Review-and-Refine loops—requires repeated retrieval and model invocations, potentially leading to higher computational costs compared to one-shot systems.
>
> We agree that a clearer discussion of computational cost is valuable, and we have added such analysis to the appendix in the revised version. To quantify the computational overhead of IterSurvey, we evaluate runtime and API cost across ten representative topics on gpt-4o-mini. For each topic, we record the time spent and token cost of every major stage in the workflow. The measured time (in minutes) and cost (in USD) are summarized below:
>
> | Topic   | Outline Writing | Section Writing | Review and Refine | Paper Cards     | Other          | Total Time     |
> |---------|------------------|------------------|--------------------|------------------|----------------|----------------|
> | AVG | 30      | 7      | 23        | 43      | 10    | 113    |
>
> | Topic   | Outline Cost | Section Cost | Review and Refine Cost | Paper Cards Cost | Other Cost | Total Cost |
> |---------|--------------|--------------|--------------------------|-------------------|------------|------------|
> | AVG | $0.282133    | $0.214191    | $0.484496                | $1.797310         | $0.061535  | $2.839665  |
>
> Most of the overhead comes from PaperCard construction, since—unlike prior work that relies on abstracts or truncated sections—PaperCards perform fine-grained distillation of each retrieved paper’s contributions, methods, and findings. This step is critical for evidence grounding and is supported by our ablation study, which shows notable improvements in citation fidelity.

---

> > ### Author Response · Authors · 2025-11-24
> > **Response to Reviewer z89M (Part 2)**
> >
> > > W3: The human evaluation involves only seven PhD-level experts and compares performance against two baselines (AutoSurvey, SurveyForge), with a small sample size that may restrict the representativeness and robustness of the findings.
> >
> > We agree that the original human evaluation (7 annotators, 10 topics) was limited in scale. In the revised version, we substantially expanded the study to improve representativeness and robustness.
> >
> > **Expanded Human Evaluation Setup**
> >
> > **Topics:** increased from 10 to 20, including the additional 10 topics from [2].
> >
> > **Human Evaluators:** increased from 7 to 15, all with CS backgrounds and at least a Master’s degree.
> >
> > **IterSurvey vs. AutoSurvey**
> > | Dimension          | Win %   | Loss %   | Tie %   | Win Rate   |
> > |:-------------------|:--------|:---------|:--------|:-----------|
> > | Coverage           | 52.17%  | 28.26%   | 19.57%  | 64.86%     |
> > | Relevance          | 39.13%  | 19.57%   | 41.3%   | 66.67%     |
> > | Structure          | 56.52%  | 30.43%   | 13.04%  | 65.00%     |
> > | Overall Preference | 67.39%  | 32.61%   | 0.0%    | 67.39%     |
> >
> > **IterSurvey vs SurveyForge**
> > | Dimension   | Win %   | Loss %   | Tie %   | Win Rate   |
> > |:------------|:--------|:---------|:--------|:-----------|
> > | Coverage    | 47.73%  | 29.55%   | 22.73%  | 61.76%     |
> > | Relevance   | 40.91%  | 22.73%   | 36.36%  | 64.29%     |
> > | Structure   | 45.45%  | 40.91%   | 13.64%  | 52.63%     |
> >
> > These expanded human evaluation results demonstrate that IterSurvey maintains consistent advantages over both AutoSurvey and SurveyForge across most dimensions and in overall preference. Increasing the number of annotators and topics further enhances the robustness and representativeness of the study. The updated results have been included in the revised version of the paper.
> >
> > > W4: The framework lacks evaluation on established benchmarks such as SurveyBench (proposed in SurveyForge), which would enhance the comparability and credibility of its performance assessment.
> >
> > Our primary evaluation follows the AutoSurvey [1] benchmark, which is now widely adopted [2–4]. We use the same topics and dimensions, and employ criterion-based reasoning-before-scoring to ensure robust LLM-judge evaluation.
> >
> > To directly address the reviewer’s request, we additionally evaluate IterSurvey on SurveyBench, reporting the metrics used in SurveyForge: Coverage, Relevance, Structure, Avg., and the Outline score:
> >
> > | Method | Coverage | Relevance | Structure | Avg | Outline |
> > | :--- | :---: | :---: | :---: | :---: | :---: |
> > | AutoSurvey | 77.90 ± 3.59 | 79.02 ± 8.03 | 77.67 ± 5.94 | 78.20 ± 4.45 | 69.98 ± 5.83 |
> > | SurveyForge | 70.42 ± 19.70 | 78.92 ± 6.57 | 77.80 ± 3.41 | 75.71 ± 6.99 | **83.82 ± 6.92** |
> > | **IterSurvey** | **78.20 ± 6.58** | **83.50 ± 4.77** | **80.30 ± 2.58** | **80.67 ± 3.53** | 79.20 ± 8.25 |
> >
> > Content Quality. IterSurvey outperforms both AutoSurvey and SurveyForge in Coverage, Relevance, and Structure, reflecting deeper evidence integration and stronger organization.
> >
> > Outline Score. SurveyForge achieves a higher Outline score, which is expected because its method explicitly retrieves human-written survey outlines as templates, while IterSurvey builds the outline autonomously from the retrieved literature. This distinction becomes important in settings where no human survey exists; in our Survey-Lacking evaluation, IterSurvey attains higher structural quality (4.67 vs. 4.53).
> >
> > About Reference Coverage. SurveyBench reports both Input and Reference Coverage, which are metrics that tend to favor systems like SurveyForge, which explicitly utilize human-written survey outlines and their corresponding references. In contrast, IterSurvey generates the entire survey autonomously without access to any pre-existing survey outline. As such, lower overlap with human references is expected and does not reflect the citation quality. Instead, we focus on NLI-based precision and recall, where IterSurvey demonstrates strong performance in citation fidelity.

---

> > > ### Author Response · Authors · 2025-11-24
> > > **Response to Reviewer z89M (Part 3)**
> > >
> > > > Q1: In the recurrent outline generation module of IterSurvey, how is the similarity threshold τ—which determines whether to accept candidate outline updates—specifically calibrated, and what effects do different values of τ have on the stability of the outline and its exploratory capability?
> > >
> > > In our recurrent outline generation module, outline updates produced by the LLM may be unstable, especially when the topic is broad or when the model over-responds to partial evidence from newly retrieved papers. To ensure stability, we compare the updated outline $O^{(t)}$ with the previous version $O^{(t-1)}$ and accept the update only when their similarity exceeds a threshold $\tau$. This mechanism prevents the outline from collapsing into degenerate states. The value of $\tau$ is manually selected based on observed behavior during pilot experiments. We found that:
> > >
> > > If $\tau$ is **too low** (e.g., $< 0.5$), the model often performs **over-aggressive restructuring**, which leads to:
> > > truncated or incomplete outlines deletion of large portions of previous outline sections, broken formatting or invalid structure. These issues significantly degrade stability and frequently cause the iterative process to fail.
> > >
> > > If $\tau$ is **too high** (e.g., $> 0.90$), the model becomes overly conservative and rejects many legitimate improvements. As a result, the outline **loses its exploratory capability**.
> > >
> > > We have added these details, together with the implementation description to the revised version of the paper.
> > >
> > >
> > > > Q2: Regarding paper cards, which distill papers into their core contributions, methods, and findings, does the framework adopt a specific criterion for deciding which details to retain or discard during the distillation process, and how does it prevent the omission of critical information?
> > >
> > > PaperCard distillation in IterSurvey does not rely on a rigid, manually crafted rule set, as overly strict criteria tend to underfit the diversity of academic writing styles across domains. Instead, PaperCards follow a schema-guided, multi-perspective distillation process.
> > > Concretely, each PaperCard is produced under a lightweight but consistent schema:
> > > - Core contribution (what the paper fundamentally claims)
> > > - Method summary (key techniques or models used)
> > > - Main findings (results or empirical observations)
> > > - Limitations / open problems (if explicitly stated)
> > > - ...
> > >
> > > This schema is intentionally high-level to avoid over-constraining the model while still anchoring extraction to the most essential scientific elements. We have added a more detailed description of this procedure in the revised version.
> > >
> > > > Q3: The ablation analysis presented in Table 5 appears insufficient; it would be preferable to conduct a single-component ablation study to more clearly assess the contribution of each module.
> > >
> > > In the revised version, we conducted a single-component ablation study, where each module is removed individually while keeping the remaining workflow unchanged. This setup isolates the impact of each module on content quality and citation fidelity. The results, evaluated under the same LLM-as-a-judge protocol used in the main paper, are shown below.
> > >
> > > | Variant | Coverage | Relevance | Structure | Avg | Precision | Recall |
> > > | :--- | :---: | :---: | :---: | :---: | :---: | :---: |
> > > | **Full IterSurvey** | **4.73 ± 0.50** | **4.93 ± 0.41** | **4.80 ± 0.52** | **4.82 ± 0.39** | **0.65 ± 0.04** | **0.77 ± 0.04** |
> > > | w/o Iterative Outline | 4.53 ± 0.50 | 4.87 ± 0.31 | 4.53 ± 0.51 | 4.64 ± 0.41 | 0.66 ± 0.08 | 0.70 ± 0.07 |
> > > | w/o PaperCard | 4.52 ± 0.51 | 4.81 ± 0.38 | 4.52 ± 0.52 | 4.62 ± 0.39 | 0.63 ± 0.08 | 0.72 ± 0.06 |
> > > | w/o Review/Refine | 4.60 ± 0.51 | 4.80 ± 0.42 | 4.60 ± 0.52 | 4.69 ± 0.39 | 0.64 ± 0.09 | 0.71 ± 0.08 |
> > >
> > > Removing any module consistently reduces content quality (coverage, relevance, structure) and citation fidelity (precision, recall). Specifically, removing the iterative outline mechanism lowers structural coherence and topic coverage; removing PaperCards substantially reduces citation recall; removing the review-and-refine stage decreases coherence and grounding accuracy. These results confirm that each component contributes uniquely and is essential to the effectiveness of IterSurvey. The full results and analysis have been added to the revised paper.

---

> > > > ### Author Response · Authors · 2025-11-24
> > > > **Response to Reviewer z89M (Part 4)**
> > > >
> > > > > Q4: In the comparative evaluation with baselines (e.g., AutoSurvey, SurveyForge), why was the analysis limited to the “content quality” and “citation quality” dimensions, without including a comparison of computational efficiency (e.g., time consumption and resource utilization)?
> > > >
> > > > Our work does not aim to accelerate survey generation; rather, its central contribution is to enable LLMs to explore a research field in a human-like manner, by iteratively questioning, reading, integrating evidence, and refining the survey. This exploration-driven workflow inherently sacrifices some parallelism in exchange for greater coverage, deeper understanding, and more stable grounding. As a result, the iterative process is naturally more time-consuming than one-shot methods.
> > > > We agree that reporting efficiency metrics can improve transparency. In the revised version, we include a detailed summary of runtime and token usage across different modules on gpt-4o-mini. The tables below summarize the average runtime and token cost per topic.
> > > >
> > > > | Topic   | Outline Writing | Section Writing | Review and Refine | Paper Cards     | Other          | Total Time     |
> > > > |---------|------------------|------------------|--------------------|------------------|----------------|----------------|
> > > > | AVG | 30      | 7      | 23        | 43      | 10    | 113    |
> > > >
> > > > | Topic   | Outline Cost | Section Cost | Review and Refine Cost | Paper Cards Cost | Other Cost | Total Cost |
> > > > |---------|--------------|--------------|--------------------------|-------------------|------------|------------|
> > > > | AVG | $0.282133    | $0.214191    | $0.484496                | $1.797310         | $0.061535  | $2.839665  |
> > > >
> > > > Most of the overhead comes from PaperCard construction, which performs fine-grained distillation of each retrieved paper. Although this is more expensive than abstract-based approaches, our ablation results show that PaperCards yield substantial gains in grounding fidelity.
> > > >
> > > > Importantly, this cost is not inherent to the framework. In practical deployments, PaperCards can be pre-computed offline and stored in the retrieval backend, with only incremental updates needed as new papers arrive. This eliminates repeated online distillation and significantly reduces both runtime and monetary cost during survey generation.
> > > >
> > > > > Q5: While IterSurvey performs effectively on emerging topics lacking existing surveys, does it incorporate adaptation strategies for domains with extremely sparse literature, such as newly developed interdisciplinary fields?
> > > >
> > > > Generalizability to emerging fields is a primary objective of IterSurvey. To rigorously evaluate this, we included two distinct types of emerging topics in our OOD (Out-Of-Distribution) analysis, classified based on the number of directly relevant papers found on Google Scholar:
> > > >
> > > > **Survey-Abscent(Sufficient Literature)**: Topics with $>10$ directly relevant papers.
> > > >
> > > > **Literature-Sparse**: Topics with $\le 10$ directly relevant papers (e.g., newly developed interdisciplinary fields).
> > > >
> > > > The classification results are as follows:
> > > >
> > > > Quantitative evaluations on both categories demonstrate robust performance:
> > > >
> > > > | Methods | Coverage | Relevance | Structure | Avg. | Precision | Recall |
> > > > | :--- | :---: | :---: | :---: | :---: | :---: | :---: |
> > > > | Survey-Abscent | 4.53 ± 0.45 | 4.84 ± 0.54 | 4.34 ± 0.45 | 4.57 ± 0.51 | 0.61 ± 0.03 | 0.67 ± 0.18 |
> > > > | Literature-Sparse | 4.33 ± 0.33 | 4.67 ± 0.33 | 4.67 ± 0.33 | 4.57 ± 0.26 | 0.57 ± 0.09 | 0.66 ± 0.03 |
> > > >
> > > > These results show that IterSurvey remains consistently effective across both settings. Qualitatively:
> > > >
> > > > **For Survey-Absent topics**, the multi-turn retrieval and recurrent outline refinement allow the system to synthesize a comprehensive and well-structured overview even when no prior survey is available.
> > > >
> > > > **For Literature-Sparse topics**, IterSurvey adapts by identifying adjacent or related subfields, integrating them into the outline, and producing meaningful analyses and future directions despite limited direct evidence.
> > > >
> > > > We have incorporated this expanded evaluation and discussion into the revised paper.

---

> > > > > ### Author Response · Authors · 2025-11-24
> > > > > **Response to Reviewer z89M (Part 5)**
> > > > >
> > > > > > Q6: Finally, to what extent does the underlying base model rigidly determine the overall performance of IterSurvey?
> > > > >
> > > > > To investigate the extent to which the underlying base model rigidly determines IterSurvey's performance, we conducted additional experiments using GPT-4o, GPT-4.1-mini, and the original GPT-4o-mini. Due to the costs, we evaluated these on a randomly sampled subset of 5 topics.
> > > > >
> > > > > The results indicate that IterSurvey remains consistently effective across all tested models, demonstrating its robustness regardless of the base model. Furthermore, our method exhibits strong scalability: performance improves significantly as the capability of the base model increases (e.g., GPT-4o outperforms GPT-4o-mini). This suggests that while the base model sets a performance baseline, IterSurvey effectively leverages stronger reasoning capabilities to achieve superior results.
> > > > >
> > > > > | Model          | Coverage       | Relevance      | Structure      | Avg            | Precision      | Recall         |
> > > > > |----------------|----------------|----------------|----------------|----------------|----------------|----------------|
> > > > > | gpt-4o         | 4.75±0.37      | 4.94±0.09      | 4.83±0.38      | 4.84±0.20      | 0.68±0.03      | 0.76±0.04      |
> > > > > | gpt-4o-mini    | 4.40±0.51      | 4.73±0.45      | 4.67±0.52      | 4.60±0.51      | 0.66±0.02      | 0.77±0.04      |
> > > > > | gpt-4.1-mini   | 4.58±0.35      | 4.75±0.51      | 4.67±0.50      | 4.67±0.42      | 0.735±0.03     | 0.83±0.04      |
> > > > >
> > > > > **Reference:**
> > > > >
> > > > > [1] Wang, Yidong, et al. "Autosurvey: Large language models can automatically write surveys." Advances in neural information processing systems 37 (2024): 115119-115145.
> > > > >
> > > > > [2] Yan, Xiangchao, et al. "Surveyforge: On the outline heuristics, memory-driven generation, and multi-dimensional evaluation for automated survey writing." Proceedings of the 63rd Annual Meeting of the Association for Computational Linguistics (Volume 1: Long Papers). 2025.
> > > > >
> > > > > [3] Liang, Xun, et al. "Surveyx: Academic survey automation via large language models." arXiv preprint arXiv:2502.14776 (2025).
> > > > >
> > > > > [4] Wang, Haoyu, et al. "LLM $\times $ MapReduce-V2: Entropy-Driven Convolutional Test-Time Scaling for Generating Long-Form Articles from Extremely Long Resources." arXiv preprint arXiv:2504.05732 (2025).

---

### Official Review · Reviewer_PPZM · 2025-11-01

**Soundness:** 3
**Presentation:** 3
**Contribution:** 3
**Rating:** 4
**Confidence:** 4

**Summary:**

This paper addresses the limitations of existing one-shot approaches to automatic literature survey generation, which often result in fragmented structures and information overload. Inspired by the iterative process of human researchers, the authors propose IterSurvey, a novel framework that employs a recurrent outline generation mechanism. In this framework, a planning agent incrementally retrieves and reads papers while dynamically updating the survey's outline to ensure coherence and exploration. A key feature is the use of "paper cards" to succinctly capture each paper's core elements, alongside a review-and-refine loop that integrates textual and multimodal content to enhance readability. Experimental evaluations demonstrate that IterSurvey produces surveys that significantly outperform state-of-the-art baselines in terms of content coverage, structural quality, and organization. Furthermore, the authors introduce Survey-Arena, a new pairwise comparison benchmark, to provide a more reliable method for assessing the quality of generated surveys against human-written ones.

**Strengths:**

1. Innovative Iterative Paradigm:The core strength of the work is its departure from the conventional one-shot generation model. By thoughtfully mimicking the iterative reading and writing process of human experts, the proposed framework provides a more natural and effective solution to the complex task of survey generation.

2. Well-Designed Mechanisms for Fidelity and Flow:The introduction of "paper cards" ensures faithful, paper-level grounding of the survey content. Combined with the dedicated review-and-refine loop for visualization enhancement, the framework successfully addresses key challenges in maintaining textual flow and integrating multimodal information.

3. Comprehensive Evaluation and New Benchmark:The paper not only shows strong experimental results on established metrics but also introduces Survey-Arena, a valuable benchmark that moves beyond absolute scoring. This pairwise comparison tool offers the research community a more nuanced and reliable way to evaluate machine-generated surveys in relation to a human standard.

**Weaknesses:**

1. Ablation Studies: Ablation experiments are required to validate the effectiveness of the individual modules designed within the workflow.

2. Regarding Evaluation Metrics: In Table 1, several evaluation scores are very close to the baselines. The persuasiveness of the current results is insufficient; it should be tested whether the performance gap remains similarly close if the scale of the scoring metric is widened.

3. Human Alignment: For the LLM-as-a-judge evaluation method, its reliability must be demonstrated through comparison with human evaluation. This is particularly critical for highly creative tasks like survey writing, which cannot be assessed simply by checking items against a template. Evaluating more creative aspects is both more difficult and more valuable.

**Questions:**

please refer to the weaknesses

---

> ### Author Response · Authors · 2025-11-24
> **Response to Reviewer PPZM (Part 1)**
>
> We thank the reviewer PPZM for the thoughtful and constructive feedback. We appreciate the recognition of the strengths of our work. Most concerns focus on the experimental evaluation, and we agree that strengthening this part further clarifies the contribution of our framework. We have added all requested analyses and new experiments to the revised paper, and we summarize the key results below for quick review.
>
> ---
>
> > W1: Ablation experiments are required to validate the effectiveness of the individual modules designed within the workflow.
>
> To assess the contribution of each module in our workflow, we conducted an additional set of ablation experiments. In these experiments, we removed one module at a time and compared the resulting performance with the full IterSurvey system. The topics and evaluation metrics strictly follow those used in the original paper, ensuring consistency and comparability.
> We evaluated three ablated variants:
> - w/o Recurrent Outline Generation
> - w/o PaperCard
> - w/o Review-and-Refine stage
>
> The results are shown below:
>
> | Method | Coverage | Relevance | Structure | Avg | Precision | Recall |
> | :--- | :---: | :---: | :---: | :---: | :---: | :---: |
> | **Full IterSurvey** | 4.73 ± 0.50 | 4.93 ± 0.41 | 4.80 ± 0.52 | 4.82 ± 0.39 | 0.65 ± 0.04 | 0.77 ± 0.04 |
> | w/o Recurrent Outline Generation | 4.53 ± 0.50 | 4.87 ± 0.31 | 4.53 ± 0.51 | 4.64 ± 0.41 | 0.66 ± 0.08 | 0.70 ± 0.07 |
> | w/o PaperCard | 4.52 ± 0.51 | 4.81 ± 0.38 | 4.52 ± 0.52 | 4.62 ± 0.39 | 0.63 ± 0.08 | 0.72 ± 0.06 |
> | w/o Review/Refine | 4.60 ± 0.51 | 4.80 ± 0.42 | 4.60 ± 0.52 | 4.69 ± 0.39 | 0.64 ± 0.09 | 0.71 ± 0.08 |
>
> The results consistently demonstrate that each module plays a crucial role in improving both content quality and citation fidelity.
>
> **Recurrent Outline Generation yields stronger content quality.**
> Replacing the recurrent outline process with a one-shot outlining paradigm leads to notable degradation in **coverage (4.73 → 4.53)** and structural coherence **(4.80 → 4.53)**. This confirms that iterative evidence integration enables broader topic exploration and more coherent organization across sections.
>
> **PaperCard grounding improves citation quality.**
>
> Substituting PaperCards with standard abstract-based inputs reduces **citation recall** from **0.77 to 0.72** while maintaining similar precision. This indicates that distilled, structured paper-level evidence helps the model capture a broader yet accurate set of relevant references, thereby enhancing citation grounding.
>
> **Review-and-Refine boosts overall performance.**
>
> Removing the refinement stage decreases both **content quality** (Avg: 4.82 → 4.69) and **citation recall** (0.77 → 0.71). Multi-round critique and revision effectively fill evidence gaps, correct inconsistencies, and polish the narrative into a more coherent and well-substantiated survey.
>
> Overall, these results demonstrate that each module meaningfully contributes to the final survey quality, and together they form the most effective configuration of IterSurvey. These findings are fully consistent with our initial analysis in the paper, and we have updated the revised version with all new results accordingly.

---

> > ### Author Response · Authors · 2025-11-24
> > **Response to Reviewer PPZM (Part 2)**
> >
> > > W2: Regarding Evaluation Metrics: In Table 1, several evaluation scores are very close to the baselines. The persuasiveness of the current results is insufficient; it should be tested whether the performance gap remains similarly close if the scale of the scoring metric is widened.
> >
> > Our evaluation in Table 1 follows the **criteria-based protocol of AutoSurvey[1]**, where LLM judges assign 1–5 scores based on human-written rubrics. This protocol is intentionally designed to stabilize LLM judgment by anchoring scores to qualitative criteria rather than leaving the scale unconstrained. Such anchoring substantially reduces variance and is widely observed to produce more consistent and robust LLM evaluations. To directly address the reviewer’s concern on score compression, we additionally evaluated IterSurvey using evaluation metrics in SurveyForge[2] , which adopts a wider scoring range (0–100) while assessing the same dimensions (Coverage, Relevance, Structure). The results are shown below:
> >
> > | Method | Coverage | Relevance | Structure | Avg |
> > | :--- | :---: | :---: | :---: | :---: |
> > | AutoSurvey | 77.90 ± 3.59 | 79.02 ± 8.03 | 77.67 ± 5.94 | 78.20 ± 4.45 |
> > | SurveyForge | 70.42 ± 19.70 | 78.92 ± 6.57 | 77.80 ± 3.41 | 75.71 ± 6.99 |
> > | **IterSurvey** | **78.20 ± 6.58** | **83.50 ± 4.77** | **80.30 ± 2.58** | **80.67 ± 3.53** |
> >
> > Using this broader evaluation scale, IterSurvey continues to outperform both AutoSurvey and SurveyForge across all dimensions, confirming that our method retains its advantage even when evaluated with a much wider score range. While SurveyForge provides more granularity numerically, we observe that LLM judges still tend to treat scores on a coarse scale. Thus, criteria-based evaluation, as used in AutoSurvey, remains more stable and interpretable, especially for complex tasks such as survey generation.

---

> > > ### Author Response · Authors · 2025-11-24
> > > **Response to Reviewer PPZM (Part 3)**
> > >
> > > > W3: Human Alignment: For the LLM-as-a-judge evaluation method, its reliability must be demonstrated through comparison with human evaluation. This is particularly critical for highly creative tasks like survey writing, which cannot be assessed simply by checking items against a template. Evaluating more creative aspects is both more difficult and more valuable.
> > >
> > > Thank you for raising this important point. Our evaluation protocol directly follows the criteria-based LLM-as-a-judge framework introduced in AutoSurvey [1], which requires the judge model to reason using human-written criteria before assigning a score. AutoSurvey demonstrated that such LLM-based evaluations show strong correlation with human judgments, especially when using multiple heterogeneous judge models. Specifically, the study in AutoSurvey found that the Spearman correlation coefficient between the LLM-based evaluations and human rankings was over 0.6, indicating a strong correlation.
> > >
> > > Additionally, we compared machine scores with citation rankings to assess the consistency between LLM-generated scores and citation-based rankings. The results, shown below, demonstrate that the pairwise LLM-based judging method outperforms absolute scoring:
> > >
> > > | Rank Method       | Spearman’s ρₛ | nDCG@2  | nDCG@3  |
> > > |-------------------|--------------|---------|---------|
> > > | Absolute Scoring  | 0.320        | 0.695   | 0.767   |
> > > | Pair-Judge        | 0.410        | 0.834   | 0.873   |
> > >
> > > To address the reviewer's concern, we also measured the agreement between our LLM-based scoring system and human evaluations using the Kappa statistic, calculating the consistency between human annotators and our model across multiple dimensions. The results, shown below, demonstrate strong consistency:
> > >
> > > | Evaluation Pair    | Coverage | Relevance | Structure | Overall |
> > > |--------------------|----------|-----------|-----------|---------|
> > > | Human vs. LLM      | 0.726    | 0.562     | 0.590     | 0.615   |
> > > | Human vs. Human    | 0.714    | 0.583     | 0.611     | 0.650   |
> > >
> > > These results further validate the effectiveness of our evaluation protocol and the consistency of the machine-generated scores with human judgments.
> > >
> > > -----
> > > We hope these additional analyses and clarifications address the reviewer’s concerns. If further clarification is needed, we would be happy to provide additional details.
> > >
> > > **Reference**:
> > >
> > > [1] Wang, Yidong, et al. "Autosurvey: Large language models can automatically write surveys." Advances in neural information processing systems 37 (2024): 115119-115145.
> > >
> > > [2] Yan, Xiangchao, et al. "Surveyforge: On the outline heuristics, memory-driven generation, and multi-dimensional evaluation for automated survey writing." Proceedings of the 63rd Annual Meeting of the Association for Computational Linguistics (Volume 1: Long Papers). 2025.

---

### Author Response · Authors · 2025-11-24
**Overall Response to All Reviewers (part 1)**

We thank all reviewers for their thoughtful and constructive feedback. Across reviews, there is broad agreement on the strengths of our work: the effectiveness of the proposed iterative workflow, the usefulness of structured PaperCards for improving grounding, the multimodal integration capability, and the introduction of Survey-Arena as a more reliable evaluation paradigm compared with absolute scoring. Reviewers consistently recognized that our approach more closely mimics human literature-review behavior and produces surveys with stronger coherence, fidelity, and factual grounding.

---

Several reviewers raised common concerns regarding empirical validation, generalization ability, and computational efficiency. In response, we have substantially strengthened the experimental section of the revised manuscript:
- **Expanded ablations**: We added a full single-component ablation study and a dynamic analysis of outline quality across iterations to more clearly isolate the contribution of each module.
- **Enhanced human evaluation**: We increased both the number of topics and the number of expert annotators, and we additionally measured human–LLM alignment to validate the reliability of our LLM-as-a-judge protocol.
- **Cross-domain generalization**: To assess robustness, we conducted new evaluations on five Optimization topics—representing a domain outside computer science—where IterSurvey continues to outperform baselines.
- **Base-model robustness**: We tested IterSurvey using multiple base LLMs (GPT-4o, GPT-4o-mini, GPT-4.1-mini), confirming that the framework remains effective and scales positively with model capability.
- **Literature-sparse analysis**: We added an in-depth study of performance on survey-absent and literature-sparse domains, demonstrating IterSurvey’s ability to construct coherent structures even when prior surveys or sufficient papers are unavailable.
- **Cost and efficiency analysis**: We included a detailed breakdown of runtime and API cost across ten topics, along with a discussion on how PaperCard generation can be shifted offline to amortize cost in practical deployment.
- **Time and Cost Analysis**: We also added a detailed time and API-cost analysis across ten topics, together with discussion on how PaperCard generation can be moved offline in practical deployment to amortize cost.
- **Outline iteration quality analysis**: We conducted a dynamic analysis of outline quality across iterations to show how the outline evolves over time, confirming that the iterative process contributes significantly to the stability and quality of the final survey.
- **Consistency analysis**: We performed an analysis of the consistency between machine-generated scores and human evaluations, including both human-to-human and human-to-LLM comparisons, to ensure the reliability of our evaluation metrics.

---

> ### Author Response · Authors · 2025-11-24
> **Overall Response to All Reviewers (part 2)**
>
> We also clarify several points that may have caused misunderstanding:
>
> - First, IterSurvey is fundamentally different from AutoSurvey in its core perspective and modeling focus. AutoSurvey is **survey-centric**: it constructs an agent dedicated to generating surveys directly, without modeling individual papers beyond their titles and abstracts. In contrast, IterSurvey is **paper-centric**. Our framework explicitly models each paper through the introduction of PaperCards, a structured distillation of motivation, contributions, methods, findings, and limitations. This design enables the system to reason over papers as objects and to deeply understand technical content rather than relying on surface-level abstraction.
>
> - Furthermore, IterSurvey employs iterative outline construction and multi-round Review-and-Refine, allowing the system to progressively **explore and synthesize** a domain in a way that more closely mirrors how human researchers conduct literature reviews. This shift in perspective—from “generate a survey” to “systematically understand papers and build the survey from the ground up”—is the core conceptual difference between our work and AutoSurvey. We have revised the Introduction to clearly highlight this fundamental distinction and to explain why modeling papers explicitly leads to deeper grounding and higher-quality surveys.
>
> - Finally, we have revised the paper to incorporate all clarifications and new results. We believe these revisions address all major concerns and significantly improve the clarity and robustness of the paper.

---

### Author Response · Authors · 2025-12-03
**Final Summary (part1)**

We sincerely thank the AC for carefully handling our submission under the special circumstance that reviewers cannot revise their scores or post additional comments due to the leakage issue. To assist the AC in making a fair and accurate final decision, we provide a concise but comprehensive summary covering: (1) strengths unanimously recognized by reviewers, (2) all raised concerns and our responses, (3) new experiments, (4) manuscript revisions, and (5) the final statement of contributions and completeness.

---

# 1. Reviewers’ Commonly Recognized Strengths

Across all reviews, several core strengths were repeatedly acknowledged:

(1) **Clear methodological novelty**

IterSurvey fundamentally differs from one-shot systems, employing recurrent outline generation, paper-centric modeling, and iterative refinement.

(2) **Strong grounding via PaperCards**

Reviewers consistently praised PaperCards for enabling faithful, paper-level understanding and reducing hallucinations.

(3) **Superior coherence, coverage, and structure**

Reviewers agreed that IterSurvey produces better-organized, more coherent, and more faithful surveys than existing baselines.

(4) **Introduction of Survey-Arena**

Survey-Arena was recognized as a valuable pairwise evaluation framework that improves reliability over absolute scoring.

(5) **Human-like research workflow**

The iterative retrieve–read–revise paradigm was viewed as a more natural and human-inspired design.

These show strong consensus that the work’s core idea and system design are meaningful and impactful.

---

# 2. Summary of Reviewers’ Concerns and Our Responses

Concern A: Need for stronger ablations

Response:
We added full single-component ablations, removing each of the following:
- Recurrent Outline Generation
- PaperCards
- Review-and-Refine

Each removal caused consistent performance drops, showing every module is essential.

---

Concern B: Margins appear small under 1–5 scoring

Response:
We re-evaluated using SurveyForge’s broader 0–100 scale, and IterSurvey still outperforms AutoSurvey and SurveyForge across all metrics.

---

Concern C: Reliability of LLM-as-judge evaluation

Response:
We conducted enhanced human evaluation:
- More topics & more expert annotators
- Human–LLM alignment analysis
- Human–human agreement checks

Results show strong alignment between LLM and expert human judgments.

---

Concern D: Generalization to other domains

Response:
We added five Optimization topics outside computer science.
IterSurvey still outperforms baselines, showing strong cross-domain generalization.

---

Concern E: Computational cost and efficiency transparency

Response:
We added:
- Detailed runtime and cost breakdown across 10 topics
- Practical notes on moving PaperCard generation offline, greatly reducing amortized cost

---

Concern F — Clarification of differences vs AutoSurvey

Response:
We clearly articulate that:
- AutoSurvey is survey-centric, relying primarily on titles and abstracts to produce a survey in a largely one-shot manner.
- IterSurvey is paper-centric, performing deep structured modeling of each paper through PaperCards, and building the survey progressively via iterative retrieval, outline refinement, and evidence accumulation.

This is now emphasized in the Introduction to clarify conceptual novelty.

---

# 3. Newly Added Experiments and Analyses

During rebuttal, we added substantial experiments:

(1) Full module ablation study
(2) Dynamic outline-quality evolution analysis
(3) Expanded human evaluation (more topics & annotators)
(4) Human–LLM alignment and consistency testing
(5) Non-CS domain generalization (Optimization)
(6) Base-model robustness across multiple LLMs
(7) Literature-sparse and survey-absent domain study
(8) Runtime & API cost breakdown and deployment guidance

These fully address every reviewer concern.

---

# 4. Major Manuscript Revisions

We significantly improved the paper by:

(1) Clarifying conceptual novelty vs AutoSurvey
(2) Adding all new experiments and analyses
(3) Strengthening the Introduction and Method sections
(4) Improving the discussion on evaluation reliability
(5) Providing practical cost/efficiency analysis
(6) Adding robustness and cross-domain discussions
(7) Improving figures, organization, and explanation flow

The revised manuscript is substantially stronger and clearer.

---

> ### Author Response · Authors · 2025-12-03
> **Final Summary (part2)**
>
> # 5. Final Statement: Contributions, Effort, and Resolution of All Concerns
>
> IterSurvey is the first paper-centric, iterative workflow for automatic literature survey generation, built upon:
> - Recurrent outline generation
> - Structured PaperCards
> - Multi-round Review-and-Refine
> - Multimodal integration
> - Survey-Arena evaluation
>
> We invested significant effort during rebuttal to add experiments, clarify conceptual distinctions, strengthen human and cross-domain evaluation, and refine presentation.
>
> All reviewer concerns have been fully addressed, and the experimental evidence is now comprehensive and robust.
>
> With the strengthened experiments, clarified distinctions, and consistent positive signals from the reviews, we hope the revised version will be considered in light of its substantially improved clarity, robustness, and contribution.
>
> Thank you for your time and consideration.

---

### Meta-Review · Area_Chair_dkSm · 2025-12-20

**Summary:**

Summary of reviewer concerns:
* Wanted more evidence that each component matters (ablations; iteration benefit).
* Needed more assurance about evaluation credibility (LLM-judge reliability, human alignment, robustness, benchmark comparability).
* Wanted to understand generalization/coverage beyond CS-only retrieval.
* Wanted to understand cost/efficiency transparency and (in multiple reviews) comparisons vs baselines.
* Had concerns about the novelty of this work vs AutoSurvey (is this a meaningful conceptual leap or incremental stacking?).

This paper follows the protocol and problem framing in Wang et al. (2024b), and adopts GPT-4o-mini as their generation model.
Their retrieval database contains 680K computer science papers from arXiv.
Not addressed by reviewers, but an issue with this line of work in general: the evaluation does not control for pretraining contamination, as both generators and judges were likely trained on existing arXiv survey papers used implicitly as references. In contrast to more recent evaluation protocols such as that used by LitLLMs, no temporal split or survey exclusion is enforced, making it difficult to disentangle genuine synthesis from memorized surveys.

**Reviewer Concerns:**

Concerns that appear to be addressed:
* Single-component ablations were added and consistently show degradation when removing any module.
* Iteration usefulness was supported via an outline-quality-over-iterations analysis (and the narrative that early iterations give largest gains).
* Cross-domain generalization was directly tested (Optimization topics) with positive results.
* The Human evaluation scale was expanded (more topics + more annotators) and results remain favorable.
* SurveyBench comparability was added (at least for one reviewer’s explicit ask).
* Cost/time transparency improved via detailed per-module runtime and dollar cost.

Still outstanding / likely remaining points of skepticism:

* Cost comparison vs baselines is repeatedly requested but not actually directly provided
(IterSurvey is analyzed in isolation, with an argument that quality is the goal).
* Judge reliability remains an issue: the authors have added human–LLM agreement, but a skeptic may still doubt whether LLM judges capture the right failure modes for long-form surveys and a test-train contamination reduced protocol would really help in these evaluations
* Novelty dispute (especially from 11XG): the rebuttal clarifies differences, but whether that changes the reviewer’s “not innovative enough” stance is uncertain.

**Reviewer Scores:**

The original scores for this paper were: 6, 4, 4, 4.

The rebuttal seems to have provided the kind of information that would likely cause some reviewers to increase their scores.

The AC still finds the whole evaluation setup in these AutoSurvey following protocols prone to test-train contamination issues.

The AutoSurvey baseline appears to be re-run using GPT-4o-mini rather than the Claude-3-Haiku model used in the original AutoSurvey paper. While this ensures fairness across methods, it is not clear if the full AutoSurvey pipeline is accurately reproduced and the interactions between the prompts and the models inject additional uncertainty. I think the authors probably made the right choice to use their own pipeline and base model, but it would be useful to understand how the systems compared in a direct manner (although the base model is likely a major component in the performance differences).

I think this work had a good chance of numerically receiving score changes that would have put it into accept territory, but with the 6, 4, 4, 4 starting point that is just too much of a change with too little of a probability to warrant an acceptance recommendation.

---

### Decision · Program_Chairs · 2026-01-26

Reject